



# The role of storm dynamics and scale in controlling urban flood response

Marie-claire ten Veldhuis[1,2], Zhengzheng Zhou[2,3,4], Long Yang[2], Shuguang Liu[3,4], and James Smith[2]

[1]Delft University of Technology, Watermanagement Department
[2]Princeton University, Hydrometeorology Group
[3]Tongji University, Department of Hydraulic Engineering, College of Civil Engineering
[4]UNEP-Tongji Institute of Environment for Sustainable Development Shanghai

*Correspondence to:* Marie-claire ten Veldhuis (j.a.e.tenveldhuis@tudelft.nl)

**Abstract.** The impact of spatial and temporal variability of rainfall on hydrological response remains poorly understood, in particular in urban catchments due to their high variability in land-use, high degree of imperviousness and the presence of stormwater infrastructure. In this study, we analyse the effect of rainfall spatial distribution with respect to basin scale and flowpath network structure on urban hydrological response based on a large, high quality observational dataset. A catalog of

279 peak events was extracted from 15 years of high resolution flow observations and radar rainfall data for five (semi)urbanised basins ranging from 7.0 to 111.1 km$^2$ in size. Results showed that largest peak flows in the event catalog were associated with storm core scales exceeding basin scale, for all except the largest basin. Spatial scale of flood-producing storm events in the smaller basins fell into two groups: storms of large spatial scales exceeding basin size or small, concentrated events, with storm core much smaller than basin size. For the majority of events, spatial rainfall variability was strongly smoothed by the flowpath

network, increasingly so for larger basin size. Correlation analysis showed that position of the storm in relation to the flowpath network was significantly correlated with peak flow in the smallest and in the two more urbanised basins. Analysis of storm movement relative to the flow path network showed that direction of storm movement, upstream or downstream relative to the flowpath network, had little influence on hydrological response variability. Slow-moving storms tend to be associated with higher peak flows and longer lag times. Unexpectedly, spatial distribution of imperviousness along the flowpath network did

not significantly alter hydrological response in relation to spatial storm characteristics. These findings show the importance of observation-based analysis in validating and improving our understanding of interactions between rainfall and catchment variability.

## 1 Introduction

The interactions between spatial and temporal variability of rainfall and hydrological response characteristics have been the

topic of numerous empirical and modelling studies in the past decades (Anquetin et al., 2010; Lobligeois et al., 2014; Morin et al., 2006; Segond et al., 2007; Syed et al., 2003; Tetzlaff and Uhlenbrook, 2005; Volpi et al., 2012; Yakir and Morin, 2011). They have shown that interactions depend on the complex interplay between rainfall variability and catchment heterogeneity in ways that remain poorly understood. This is the case in particular for urban catchments where high variability in land-use,





high degree of imperviousness and the presence of stormwater drainage and detention infrastructure increase the complexity of hydrological response (e.g., Bruni et al., 2015; Fletcher et al., 2013; Meierdiercks et al., 2010; Smith et al., 2005, 2013a; Yang et al., 2016).

Urbanisation tends to be associated with higher peak flows induced by reduced infiltration rates on impervious surfaces and

with shorter response times. (e.g., Rose and Peters, 2001; Cheng and Wang, 2002; Du et al., 2012; Huang et al., 2008). On the other hand, several studies have found mixed effects of urbanisation on peak flows and response times, associated with a combination of imperviousness and flood mitigation measures, especially for basins where urbanisation has predominantly taken place after implementation of stormwater control legistlation (e.g., Smith et al., 2013a; Hopkins et al., 2015; Miller et al., 2014). Niemczynowicz (1999); Schilling (1991) pointed out the importance of spatially distributed rainfall information at high

resolution to study response in urban basins. Thanks to the advances of weather radar, such information is becoming increasingly available (Krajewski and Smith, 2002; Berne and Krajewski, 2013), typically at 1 km spatial resolution (Smith et al., 2007), and in some cases down to less than 100 m (Otto and Russchenberg, 2011; Chen and Chandrasekar, 2015). Wright et al. (2014b) analysed flow variability in three semi-urbanised catchments in relation to different radar rainfall products and found that storm event water balance and hydrological response times varied with the radar product used for analysis. Berne et al.

(2004) derived relationships for critical rainfall resolution for urban hydrology, using high resolution radar rainfall datasets over 6 basins in the Mediterranean region. They found that temporal and spatial rainfall resolution required for urban hydrological analysis varied from about 5 min, 3 km for basins $\sim$10 km$^2$, to about 3 min, 2 km for basins of $\sim$1 km$^2$ scale. Radar rainfall data have been used in various studies in recent decades to drive hydrological models and sensitivity of urban hydrological response to spatial and temporal rainfall variability. Bruni et al. (2015) and Ochoa-Rodriguez et al. (2015) used rainfall data

from a polarimetric rainfall radar, at $\sim$30-100 meters and minute resolution to drive semi-distributed hydrodynamic models of one respectively seven highly urbanised catchments in NW-Europe to study urban hydrological response for a range of rainfall input resolutions. They found that sensitivity of flows to rainfall variability increased for smaller basin sizes and that hydrological response was more sensitive to change in temporal than in spatial rainfall input resolution. Gires et al. (2012) quantified the impact of unmeasured small scale rainfall variability on urban runoff for an urban catchment in London, by downscaling

radar rainfall data from 1 km and 5 min resolution to a resolution 9-8 times higher in space and 4-1 times higher in time. Uncertainty in simulated peak flow associated with small-scale rainfall variability reached 25% and 40% respectively for frontal and convective events. Rafieeinasab et al. (2015) analysed sensitivity of hydrological response to rainfall variability for 5 urban catchments of different sizes, located in the City of Arlington and Grand Prairie (U.S.), using a distributed hydrological model. They found that while flow variability was better captured using higher resolution rainfall input, errors in reproducing flow by

the models remained equally large, with peak flow over- and underestimations by more than 100%.

Wright et al. (2014a) analysed hydrological response for 4 semi-urbanised basins in Charlotte watershed, North Carolina, using a Gridded Surface Subsurface Hydrologic Analysis (GSSHA) model to examine the effect of rainfall time and length scales on flood response. They found that peak flows in the larger basins ($\sim$50-100 km$^2$) were dominated by large-scale storms, while more concentrated organized thunderstorm systems dominated in the smaller basins ($\sim$7-30 km$^2$). They also identified limita-

tions of this and similar modelling studies, where hydrologic response may be attributable to errors in radar rainfall estimates





or to features that were omitted or poorly represented in the model, such as detention ponds, the spatial distribution of layered soils, and, in particular, initial soil moisture.

Smith et al. (2002) used a data-driven approach to study relationships between temporal and spatial rainfall variability and hydrological response in urban basins. They introduced the concept of rainfall-weighted flow distance, representing storm position and movement relative to the flowpath network in the basin. In their study, they analysed hydrological response in five semi-urbanised basins in the US for five extreme flood-producing storms based on detailed radar rainfall and flow observation datasets. They found that fractional coverage of a basin by heavy rainfall is a key element of scale-dependent flood response: storm event scales, i.e. spatial (area, length) and temporal (duration) smaller than the basin scale (basins length, response time) leads to lower runoff ratios and flood peak as compared to when scales of rainfall and basin are similar. Storm motion was found to be amplifying peak flow under particular conditions: storm motion from the lower basin to the upper basin on a timescale of approximately 2 hours served to amplify peak discharge for the case of a large, $\sim$100 km$^2$ basin, relative to other modes of storm motion. In Smith et al. (2005), spatial rainfall variability in relation to the flowpath network was analysed for 25 flash flood producing storms in a 14 km$^2$ urban watershed. They found that spatial rainfall variability was strongly smoothed by the flowpath network resulting in hydrological response for storms with widely varying spatial rainfall variability being strikingly similar.

Other authors have used similar concepts to study hydrological response in natural basins. In an extensive study of 300 events over a 148 km$^2$ basin in Arizona, Syed et al. (2003) found that runoff volume and peak were strongly correlated with areal coverage by the storm core (>25 mm/h rainfall intensity). The importance of storm core's position increased with basin size, with storm core positioned in the central portion of the watershed producing more runoff and higher flood peaks. Morin et al. (2006) found that the sensitivity of flood response (in terms of flood peak magnitude and peak timing) to spatial rainfall variability increased with storm intensity, which they attributed to high flow velocities during intense storms. Similar results were found by Lobligeois et al. (2014) who analysed the influence of spatial rainfall variability on hydrological response in 181 catchments in France based on spatial rainfall variability, storm position and catchment-scale storm velocity indices. They found that flow simulations by hydrological models benefited from spatially distributed rainfall input for large catchments and strongly spatially distributed rainfall fields. Nicotina et al. (2008) analysed rainfall variability in a numerical study for large basins up to several thousand km$^2$ and found that spatial variability of a storm was more important than variability in total rainfall volume over the basins. This was attributed to the dominant influence of hillslope flow at scales typically smaller than the rainfall variability scale, smoothing differences in travel times to the basin outlet. Only in very large basins (>8000 km$^2$) channel flow became more important, leading to stronger sensitivity to rainfall spatial distribution. Zoccatelli et al. (2011) analysed rainfall coverage, storm position and movement relative to the flowpath network for 5 storms in 5 different basins in south-east Europe. Based on a model sensitivity study, they found that peak timing error introduced by neglecting rainfall spatial variability ranged between 30 % to 72% of the corresponding catchment response time. Nikolopoulos et al. (2014) analysed the role of storm motion using radar rainfall data to drive two models of varying complexity. They found that storm motion did not play a significant role in generating hydrologic response for a large storm event, in basins sized 8-623 km$^2$.





Emmanuel et al. (2015) investigated impacts or rainfall spatial variability on hydrological response using a model simulation approach and found significant dispersion in results obtained for events for different simulation scenarios, showing the need for studying larger sets of events to derive robust general conclusions. Modelling studies reported in the literature have remained unconclusive with respect to the interactions between rainfall and catchment scales (Ogden et al., 2011; Morin et al., 2006;

Nicotina et al., 2008; Rafieeinasab et al., 2015). This emphasises the importance of using field observations to corroborate preliminary conclusions drawn from model simulations.

In this study, we extracted a catalog of 279 flood events from 15 years of high quality flow observations and in five nested (semi-)urbanised basins in Charlotte region, North Carolina (US). By flood events we understand the set of events associated

with the top five largest peak flows per year. Observational resources for the Charlotte metropolitan region are exceptionally rich (e.g., Smith et al., 2002; Wright et al., 2013). The region is covered by two National Weather Service WSR-88D (Weather Surveillance Radar-1988 Doppler) radars, both of which were deployed in 1995. A dense network of rain gauges and stream gauges was installed by the U.S. Geological Survey (USGS) in 1995. We analyse the influence of spatial scale, position and movement of storms relative to the flow path network as well as interactions with spatial distribution of imperviousness on

urban flood response. We aimed to address the following questions:

– How does rainfall scale interact with basin scale in determining urban flood response? We use fractional coverage to express the relation between rainfall scale versus basin scale and to investigate the dependencies of flood peak magnitude and lag time on rainfall scale.

– Does the position of a storm in relation to the flow path network influence flood response? We use the concept or rainfall-
weighted flow distance (RWD) to identify the position of a storm relative to the flowpath network and analyse whether storms concentrated in the upstream part of the catchment are associated with significantly different response compared to storms concentrated in the centre or near the basin outlet.

– How does storm direction and velocity in relation to the flow path network influence flood response? We use first-order differences in RWD to characterise storm movement and investigate if storms passing over the basin in downstream
direction lead to significantly different hydrological response compared to storms moving in upstream direction and storms moving perpendicular to the main flow direction.

– How does the position of a storm in relation to the spatial distribution of imperviousness influence flow response?

This paper is organised as follows: in section 2, the case study area, datasets and methods used in this study are introduced. Results are presented and discussed in section 3, followed by summary and conclusions in section 4.



## 2   Data and Methods

### 2.1   Study region, rainfall and flow datasets

The data used in the study were collected at five USGS stream gauging stations in Charlotte-Mecklenburg county, North Carolina. Gauging stations are located at the outlet of hydrological basins that range from 7.0 km$^2$ to 111.1 km$^2$ in size. The area is
largely covered by low to high intensity urban development, covering 60% to 100% of basin areas. Percentage impervious cover varies from 25% in the least developed to 48% in the most urbanised basin covering the city centre of Charlotte. Figure 1 shows a map with the location of the area, catchment boundaries and location of stream gauges used in the analysis. High-resolution (30 m) gridded datasets were used for terrain elevation (National Map of USGS, http://viewer.nationalmap.gov/), impervious cover and land-use/land-cover (LULC, from National Land Cover Dataset NLCD, available at http://www.mrlc.gov/).

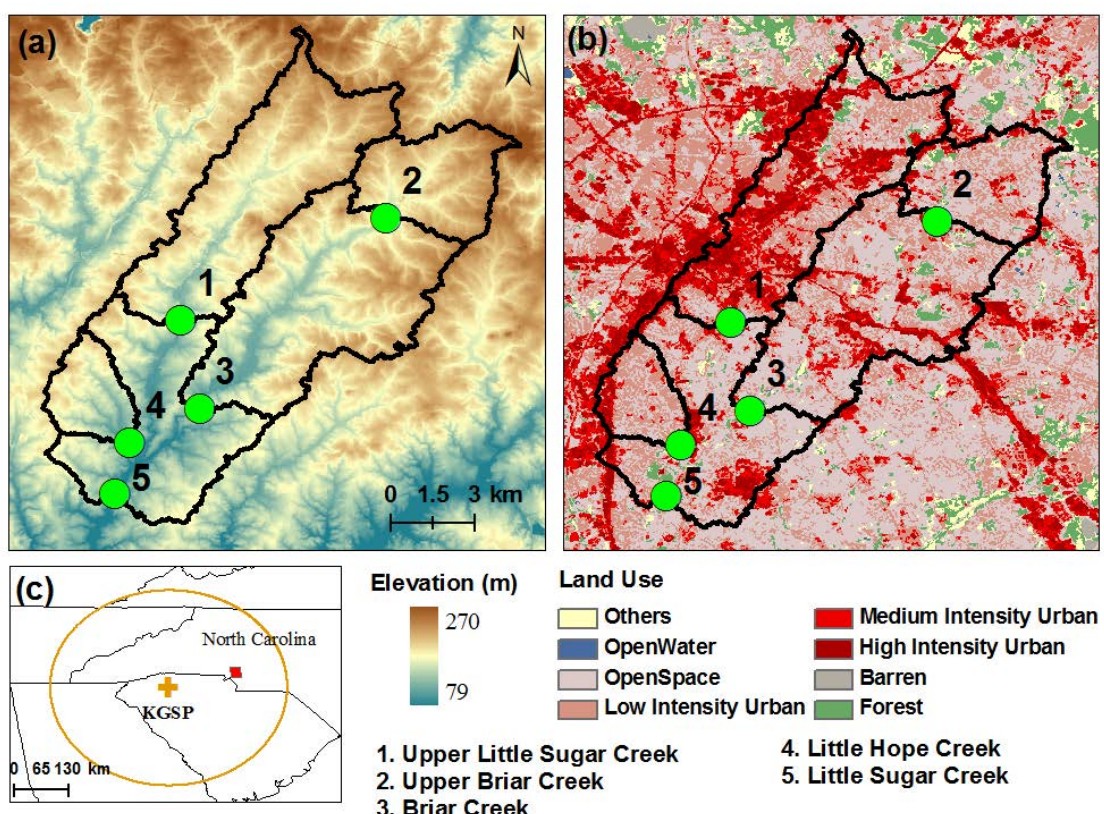

**Figure 1.** Location of Little Sugar Creek catchment (c), topography (a), landuse/landcover (b), location and boundaries of subbasins, including locations of flow gauges, location of rainfall radar





The focus of this was Little Sugar Creek catchment, upstream of the flow gauge at Archdale, with a total drainage area of 111 km$^2$. Additionally, we used data from basin nested within the main basin, sized 7.0, 13.3, 31.5 and 48.5 km$^2$. Stream gage data were collected at 5 to 15 minute intervals over the period 2001-2015. For this study, all flow data were linearly interpolated 1-minute values and converted to time zone in UTC. Gauges measure water depth using pressure transducers, accuracy standard

set by the USGS Office of Surface Water for stage measurement is approximately 0.01 foot (ft) or 0.2 percent of the effective stage. Flows are derived from stage-discharge curves that were established based on protocols developed by USGS and include manual flow measurements during site visits performed by USGS staff. As part of this procedure, stage-discharge curves are checked and recalibrated during site visits several times per year. More information on gauge data and field measurements is available at http://waterdata.usgs.gov/nc/nwis. Flow datasets for the Charlotte region are of exceptionally high quality and

consistency as data collection protocol and gauge locations have remained unchanged over decades.

A summary of basin characteristics in Little Sugar Creek catchment is provided in table 1. (Sub-)basin areas range from 7.0 to 111.1 km$^2$, impervious cover from 23.9 to 48.2%, urban land-use (excl. parks and lawns) covers 47.1 to 79.1% of the basin area. Upper Little Sugar (Upper LSugar hereafter) is the most urbanised basin, covered by the urban core of the city of Charlotte. Upper and lower Briar (hereafter Upper Briar and Lower Briar) are the least urbanised basins, with impervious

cover of 23.9 and 24.7% respectively; Little Hope (LHope) is the smallest basin in size. Maximum flow distance along the flowpath network varies from 49 km for the smallest to 213 km for the largest basin. Basin compactness, computed as the ratio of basin area over perimeter squared, is highest for Little Hope and lowest for Upper LSugar, showing that the latter is the most elongated basin. Dams have been implemented in three of the basins, all for recreational purposes, according to the National Inventory of Dams (nid.usace.army.mil/cm_apex). Storage volume varies from approximately 0.1 to 2 mm (dam

storage volume divided by basin area).

Based on data from the USGS flow datasets, we established a catalog of flood events, based on "peak-over-threshold" selection such that we have, on average, five events per year over the period 2001-2015. Since radar rainfall data were only available for the summer season, April to September, events were extracted exclusively for this period. Flood events are local maxima in discharge for which there is not a larger discharge in a time window of 12 hours centred on the peak time. Events

with incomplete rainfall or discharge data were excluded from the dataset. This resulted in a catalog of 50 to 69 storm events per basin (see table 1).

Rainfall amounts were computed for the time period associated with each of the flood events, based on radar rainfall data. Fifteen years (2001-2015) of high-resolution (15 min, 1 km$^2$) Hydro-NEXRAD radar rainfall fields were available for this study, based on volume scan reflectivity observations from the NWS-operated Weather Surveillance Radar 1988 Doppler

(WSR-88D) radar in Greer, South Carolina (radar code KGSP, see figure 1c). The Hydro-NEXRAD processing system was developed to generate radar rainfall estimates for hydrologic applications by converting three-dimensional polar-coordinate volume scan reflectivity fields from NWS WSR- 88D radars into two-dimensional Cartesian surface rainfall fields (Krajewski et al., 2011). The standard convective rainfall-reflectivity (Z-R) relationship ($R = aZ^b$, where a=0.017, b=0.714; R is rain rate in mm/h, Z is radar reflectivity in mm$^6$/m$^3$), a 53 dBZ hail threshold, and several standard quality control algorithms are used

(see Seo et al. (2011) for more details). No range correction algorithms are used in this study. The data set has been extensively



**Table 1.** Summary of hydrological basins in the Little Sugar Creek catchment: basin area [km2], imperviousness [%], slope [-], land use coverage (high intensity, medium intensity, low intensity urban development) [%], maximum flow distance [km], number of dams regulating stormwater flows [-], number of POT flood events used for analysis [-]

| Name | USGS ID | Drainage area (km$^2$) | Slope (-) | Max flow distance (km) (-) | Basin compactness (%) | Imperviousness | Land use coverage (%) | | | Nr of dams (-) | Nr of events (-) |
|---|---|---|---|---|---|---|---|---|---|---|---|
| | | | | | | | high int | med int | low int | | |
| Little Hope | 02146470 | 7.0 | 2.2 | 49 | 2.6 | 32.2 | 9.3 | 9.4 | 48.5 | 0 | 54 |
| Upper Briar | 0214642825 | 13.3 | 1.9 | 58 | 2.3 | 23.9 | 3.6 | 9.3 | 34.2 | 1 | 50 |
| Upper Little Sugar | 02146409 | 31.5 | 2.2 | 128 | 1.4 | 48.2 | 22.5 | 24 | 32.6 | 0 | 69 |
| Lower Briar | 0214645022 | 48.5 | 2.4 | 168 | 1.6 | 24.7 | 4.5 | 9.9 | 32.8 | 5 | 54 |
| Lower Little Sugar | 02146507 | 111.1 | 2.4 | 213 | 1.6 | 32.0 | 10.3 | 14.1 | 32.8 | 8 | 52 |

validated in Wright et al. (2014b) and used for rainfall frequency analysis in Wright et al. (2013). Mean field bias correction of the radar rainfall is done at the daily scale using 71 rain gages from the Charlotte Rain gauge Network (CRN) (see Wright et al. (2014b)). Radar-based rainfall estimates captured variability of rainfall at time scales of 5-15 minutes and space scales of 1 km$^2$. We used rainfall data at a temporal resolution of 15 minutes to avoid sensitivity to sampling error at the 5 minute

5   time-scale. Radar rainfall data were spatially resampled at 30 meters resolution using inverse-distance interpolation between radar pixel centroids, to enable computation of rainfall redistribution relative to the flow path network (as will be explained in the next section). Basin-average rainfall rates were also computed, based on spatial aggregation of rainfall values over 1 km$^2$ pixels within the catchment boundaries of the individual basins (percent of each 1 km$^2$ grid in the basin was computed for pixels overlapping catchment boundaries).

10   ## 2.2   Methods

### 2.2.1   Hydrograph and basin average rainfall characteristics

The following rainfall metrics were defined per event, based on basin-average rainfall rates derived from radar-rainfall data at 15 minutes, 1 km$^2$ resolution:

Basin-average rainfall rate:

15   $$R_b(t) = \int_0^T R(x,t)dx \qquad (1)$$





Where: $R_b(t)$: basin-average rainfall rate at times $t$ (mm/h); $R(x,t)$: rain rate at pixel $x$ (1x1 km$^2$), at time $t$ (time step is 15 minutes); $T$: time period of selected event, from 12 hours before the maximum peak flow of a storm event until 12 hours after the peak:

$$T = \int_0^T I(R_b(x,t) > 0)dt, \quad \text{Where:} \quad I(R_b(x,t) > 0) = \begin{cases} 1 \; for \; R_b(x,t) > 0 \\ 0 \quad \text{otherwise} \end{cases} \tag{2}$$

Total rainfall depth per event:

$$R_{b,tot} = \int_0^T R_b(t) \tag{3}$$

Maximum 15-minute rainfall intensity:

$$R_{b,max} = \max\{R_b(t) : t \in [0, T] \tag{4}$$

The following metrics were used to analyse relationships between rainfall and hydrologic response; flow values were nor-
malised by basin area and expressed in m$^3$/s/km$^2$, to allow comparison among different basins:

Maximum normalised peak flow:

$$Q_{max} = max\{Q(t)A^{-1} : t \in [0, T] \tag{5}$$

Where: Q: instantaneous flow observation, at 1 minute intervals; A: basin area

Total normalised runoff volume:

$$Q_{tot} = \int_0^T Q(t)A^{-1} \, dt \tag{6}$$

Flood event duration: $T_Q$, defined as the interval between the time when the unit hydrograph continuously rises above 0.05 m$^3$/s/km$^2$ and falls below 0.01 m$^3$/s/km$^2$. Thresholds were established based on visual inspection of the hydrographs and work well for flood events with a single peak (or events separated from other flood peaks by at least 6 hours). For flood events with multiple peaks (i.e. flood peaks that are either preceded or followed by another flood peak within a short time, e.g., 1 hour),
these thresholds can result in anomalously long event durations that are not representative of hydrological response behaviour. For these events, we manually determined the start and end time for each of the "multi-peak" events by visually inspecting the hydrographs. We further checked the duration for "single-peak" events through visual inspections, to ensure consistency in the definition of event duration.

Lag time ($T_l$): time difference between basin-average rainfall peak and maximum peak flow, computed from the time distance
between the time of peak flow and time of basin-average maximum rainfall intensity during the preceding 12-hour time period. In our initial analyses, we used two methods to compute lag times, based on peak-to-peak and on distance between centroids of hyetograph and hydrograph. The latter resulted in a large number of negative lag time values, associated with events with





multiple rainfall and/or peak flows. After visual inspection of hyetographs and hydrograph peaks, we decided that peak-to-peak time gave a better representation of the response between rainfall and peak flows for most events, hence we decided to stick to this lag time definition in our analyses.

Runoff ratio: normalised runoff divided by total basin-average rainfall over the duration of the flood event ($T_Q$)

Peak ratio: normalised peak flow (flow divided by basin area) divided by rainfall peak intensity

### 2.2.2   Rainfall spatial characteristics: spatial variability, fractional coverage and rainfall-weighted flow distance

We used fractional coverage of the basin by rainfall above a given threshold to analyse the influence of rainfall scale in relation to basin scale on hydrological response. Additionally we used the concept of rainfall-weighted flow distance (RWD), as first introduced by Smith et al. (2002). RWD provides a representation of rainfall variability relative to a distance metric imposed

by the flow path network (Smith et al., 2005). The distance function $\{d(x); x \in A\}$ is the flow distance from point $x$ within the basin to the outlet of the basin.

Rainfall fractional coverage was computed as follows:

$$R_c(t) = max\{\frac{1}{A} \int\limits_A I(R(x,t))dx\} \tag{7}$$

Where: $I(R(x,t))$ is the indicator function, and equals 1 when $R(x,t) \geqslant r$ or 0 otherwise; $R_c(t)$: maximum portion of basin area receiving rainfall equal to or exceeding $r$ mm/h rainfall. We used a threshold of $r = 25$mm/h, representative of high inten-

sity rainfall, likely to be associated with extreme peak flows.

RWD is normalised by the maximum flow distance in the network and is defined as follows:

$$D(t) = \frac{1}{d_{max}} \int\limits_A w(t,x)d(x)dx \tag{8}$$

where: $D(t)$: RWD normalised by maximum flow distance (-); $d_{max} = \{d(x); x \in A\}$, maximum flow distance in the flow

path network (m) and

$$w(t,x) = \frac{r(t,x)}{\int_A r(t,x)dx} \tag{9}$$

The random variable $D(t)$ takes values from 0 to 1: low values of $D_{Rw}(t)$ are associated with rainfall that is spatially concentrated near the outlet, high values with rainfall concentrated near the headwaters of the basin. For uniformly distributed rainfall, all weights across the basin are equal and $D_{Rw}(t)$ represents the mean flow distance imposed by the flow path network:

$$\bar{d} = \int\limits_A d(x)dx \tag{10}$$

RWDs were computed per time step as well as for the total accumulated rainfall per storm event. The first provides information on storm movement over the basin in relation to the flow path network and combines both temporal and spatial rainfall





variation (Smith et al., 2002), while the latter focuses on the spatial aspect of rainfall distribution, summarising it for the total accumulated rainfall per storm event (Smith et al., 2005).

RWD dispersion was computed, to provide an indication of whether rainfall spatial distribution as imposed by the flowpath network is unimodal or multimodal. The normalised RWD dispersion was defined as (Smith et al., 2005):

$$S(t) = \frac{1}{\bar{s}} \left\{ \int_A w(t,x)[d(x) - \bar{d}]^2 dx \right\}^{\frac{1}{2}} \tag{11}$$

Where $\bar{s}$ is the dispersion for uniform rainfall:

$$\bar{s} = \left\{ \int_A [d(x) - \bar{d}]^2 dx \right\}^{\frac{1}{2}} \tag{12}$$

RWD dispersion takes the value 1 for uniform rainfall; values below 1 are associated with unimodal spatially distributed rainfall and values above 1 represent multimodal spatially distributed rainfall peaks in relation to the flowpath network. To further investigate the influence of spatial distribution of urbanisation on urban flood response, we computed RWD strictly for pixels with impervious cover larger than 80%, classified as high-intensity development in the NLCD dataset.

$$D_I(t) = \frac{1}{d_{max}} \int_A I(x)w(t,x)d(x)dx \tag{13}$$

Where $I(x)$ is an impervious indicator and takes value 1 for pixels with impervious cover > 80% and 0 for pixels with impervious cover < 80%.

### 2.2.3 Summary statistics and correlation analysis

Metrics associated with RWD are sensitive to the length of the time window over which they are computed (Smith et al., 2002); Nikolopoulos et al. (2014). We used a range of time windows of $x$ hour rainfall, $x$ varying from 0.5 to 3 hours; corresponding to the time scales of storm duration and lag time for the largest two basins (median storm durations 3 and 3.5 hours, median lag times 1.7 and 2.0 hours respectively). The time window was centered over the time of event-maximum rainfall intensity. The following summary statistics were retained for RWD: mean, minimum, maximum, coefficient of variation and gradient as well as RWD for event-total accumulated rainfall. We analysed time-varying spatial coverage by the storm core (>25 mm/h), $\Delta Rcov/\Delta t$, in relation to basin-average rainfall $\Delta R/\Delta t$ to see how much of change in rainfall intensity is associated with change in storm core coverage of the basin. We analysed $\Delta R/\Delta t$ versus $\Delta RWD/\Delta t$ to see how change in rainfall intensity relates to movement of the storm relative to the flow path network. Correlation analyses were performed for all combinations of metrics associated with basin-average rainfall, flow hydrograph, spatial rainfall variability and imperviousness distribution, based on Spearman rank correlations. Correlations were tested for significance at the 5% level (p-value < 0.05, based on t-test).





## 3 Results and discussion

### 3.1 Rainfall and hydrograph characteristics of the selected events

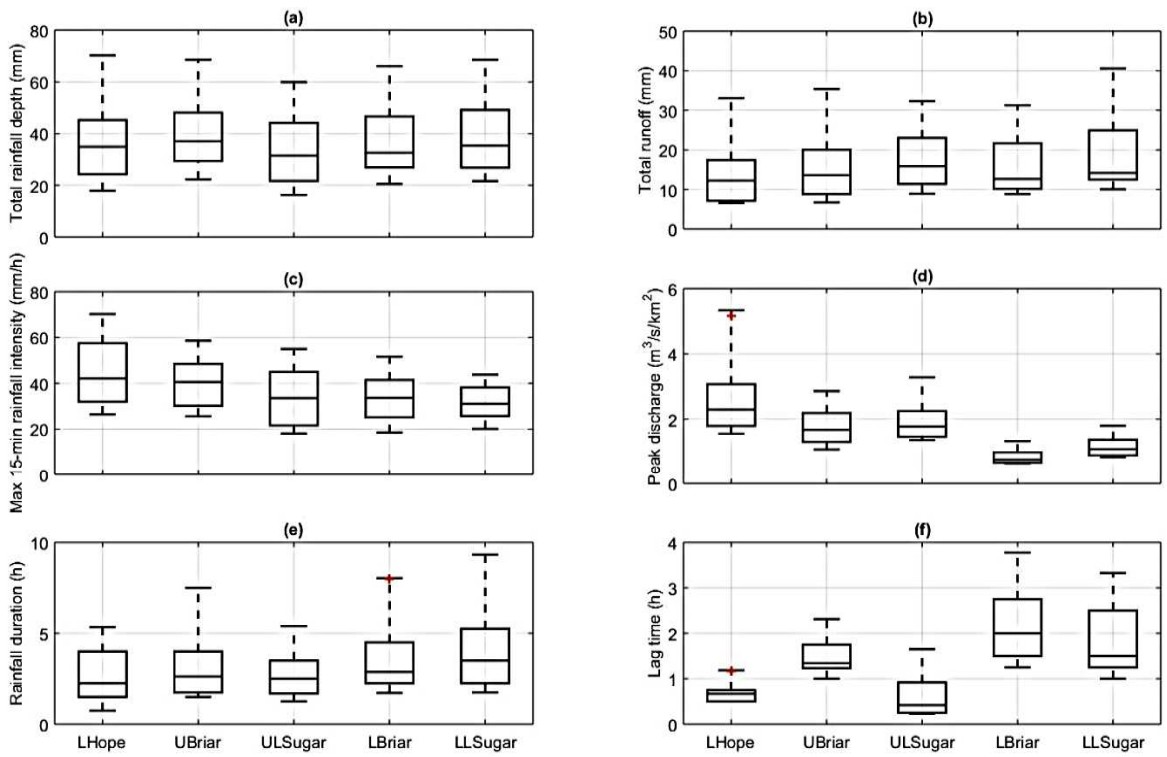

**Figure 2.** Boxplots showing 10%, 25%, 50%, 75% and 90% quantiles of characteristic rainfall and flow values for all events, per basin: Total basin-average rainfall depth (**a**), total normalised runoff volume in mm (**b**), max 15-min rainfall intensities in mm/h (**c**), normalised peak flows in m$^3$/s/km$^2$ (**d**), rainfall duration in hours (**e**), lag time (**f**)

Distributions of characteristics for rainfall and flow hydrographs for the catalog of selected events are visualised in figure 2. Basin-average rainfall depth was of the same order of magnitude for all basins, median values varying between 32.2 and 37.0 mm. Runoff volumes are slightly lower for the smallest two basins in terms of their median values and less skewed. Peak rainfall intensities show stronger variation with basin size: median for peak 15-minute rainfall intensity decreases from 41.7 mm/h for the smallest to 31.2 mm/h for the largest basin. Peak rainfall intensity varied by factor of 10 approximately across the set of selected peak events per basin (9.5 to 87.6 mm/h for Lower LSugar; 9.3 to 83.2 mm/h for Lower Briar; 9.7 to 91.7 mm/h for Upper LSugar; 8.8 to 90.7 mm/h for Upper Briar; 10.4 to 118.5 mm/h for LHope). Figure 2 (**d**) shows large differences in peak flow variability between the basins. Lower Briar had lowest median normalised peak flows and smallest variability in



peak flows, tied to a combination of large area size and low impervious cover compared to other basins. The smallest basin, LHope, has a strongly skewed peak flow distribution, with highest median as well as largest variability in normalised peak flow values compared to the other basins. Upper LSugar, the most impervious basin shows a high median peak flow value relative to its basin size. Flow variability is low for Upper and Lower LSugar in relation to their basin size, in terms of CV values: 0.37

and 0. 46 for Upper and Lower LSugar; 0.65, 0.46 and 0.44 for LHope, Upper and Lower Briar. Similar results were found for a wider range of basins in this region in ten Veldhuis and Schleiss (accepted for publication), who concluded that for the basins in the Charlotte catchment, flow regulation and peak flow restrictions induced by capacity constraints result in an overall effect of peak flow reduction associated with urbanisation.

Rainfall duration varied from approximately 0.5 to 14 hours, representing a wide range from concentrated, single peak events

to prolonged, multi-peak events 2 (**e**). Distributions show a large variability and highly skewed; high percentiles were mainly associated with storm events with multiple rainfall peaks. Lag times (figure 2**f**), computed as time between maximum rainfall intensity and peak flow, are strongly tied to a combination of basin area size and impervious cover. Upper LSugar, the most urbanised basin, has the shortest median lag time in combination with a high variability in lag times between storm events. The two largest basins have lag times between 1 and 4 hours (10-90% range). Lower LSugar has a shorter median lag time than

Lower Briar, despite its larger size. This confirms findings in an earlier study by Smith et al. (2002), who found that peaks at Lower LSugar are mostly linked to discharge from the highly urbanised Upper LSugar basin. Variability in lag time per basin was of similar order of magnitude as lag time variability between basins. Outliers in lag time were generally associated with multi-peak events, where multiple rainfall peaks caused one or more peak flows over a prolonged period of time. Runoff ratios vary mainly with imperviousness degree: largest median runoff ratio was found for Upper LSugar (0.51), followed by Lower

LSugar (0.44), Lower Briar (0.38) and the two smallest basins, Upper Briar (0.35) and LHope (0.34). Variability in runoff ratio, expressed in terms of coefficient of variation, is low for Upper and Lower LSugar basins compared to the other basins (figure not shown). This effect is even stronger for peak-to-peak ratios: variability in terms of coefficient of variation is very low for the more impervious basins (0.5 and 0.6 respectively for Upper and Lower LSugar) compared to the other basins (CV-values 5.1, 4.2 and 3.7 for LHope, Upper and Lower Briar, respectively).

## 3.2   Rainfall spatial variability and fractional basin coverage

Spatial rainfall variability was analysed based on coefficient of variation (CV) of rainfall intensities per time step. Mean CV values vary from 1.24 for the smallest to 3.51 for the largest basin, showing that rainfall tends to be more spatially uniform for smaller basins compared to larger basins. Spatial variability is high compared to temporal rainfall variability based on basin-average rainfall, where CV values vary between 0.94 and 1.03 (no clear relation with basin size). This is partially a result of the

difference in aggregation scales: basin-average rainfall is aggregated over 7 to 111 km$^2$ and 15 minutes, while spatially variable rainfall is aggregated over 1 km$^2$ and several hours rainfall duration. Additionally, spatially varied rainfall data include far more zero values, which leads to strongly skewed distributions, as is confirmed by large differences between mean and median, while these differences are small for temporal rainfall variability. Still, these results show that rainfall for the selected flood events tends to be highly spatially variable. Moreover, spatial variability changes with time, more strongly so for the larger than for the



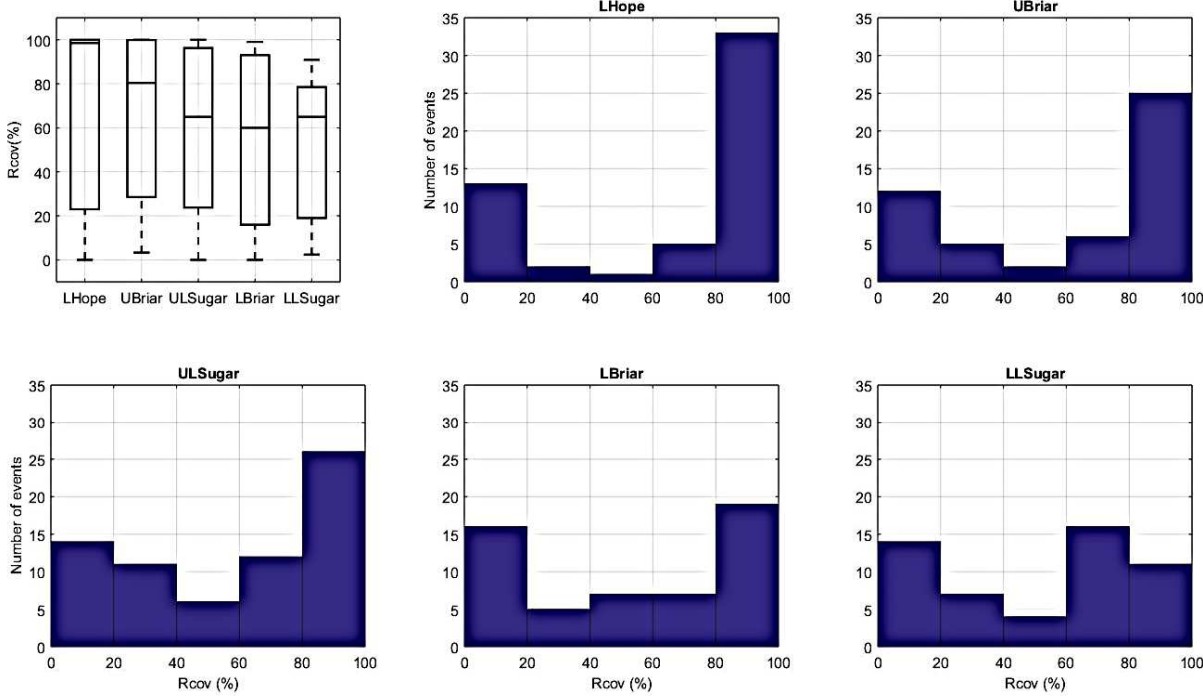

**Figure 3.** Boxplots showing 10%, 25%, 50%, 75% and 90% quantiles (**a**) and empirical histograms (**b**) of fractional basin coverage by maximum rainfall intensities >25 mm/h, representative of the storm core, for the five basins in the Little Sugar Creek catchment

smaller basins. This is a characteristic of hydroclimatic conditions in this region north-east of the Appalachians, as confirmed for instance by Zhou et al. (in review) . Similar results were found by Lobligeois et al. (2014), who analysed spatial variability of storm events associated with the largest 20 flood events in 181 basins in France. They showed that spatial rainfall variability was strongly dependent on hydroclimatic regions, with high variability occurring in the Mediterranean area, associated with
5   summer convective storms, and low variability over much of the northern and western regions of France.

Figure 3 shows boxplots and emprical histograms of fractional rainfall coverage, i.e. the maximum percentage of basin area covered by rainfall intensities larger than 25 mm/h during storm events, representing the most intense core of the storm. The



**Table 2.** Overlap in top flood producing storms for the five basins in Little Sugar Creek catchment

| Basin name | LLSugar | LBriar | ULSugar | UBriar | LHope |
|---|---|---|---|---|---|
| LLSugar | 52 | 36 | 36 | 32 | 28 |
| LBriar |  | 54 | 30 | 32 | 21 |
| ULSugar |  |  | 69 | 30 | 34 |
| UBriar |  |  |  | 50 | 20 |
| LHope |  |  |  |  | 54 |

boxplots show that storm cores exceed basin scale for 43% and 23% of the storms in the two smallest basins (7 and 13.3 km$^2$, respectively). For the larger basins this decreases to 10, 4 and 2% respectively (for basin size 31.5, 48.5 and 111.1 km$^2$). Similar results were shown by Smith et al. (2002) and Syed et al. (2003) for the same range of (sub)basin sizes, for respectively 5 storms using radar rainfall data and for 300 summer storms in Arizona using interpolated rain gauge data. Another interesting features appears in the empirical histograms: for the smaller basins fractional coverage values tends to be either small compared to basin size (coverage 0-20%) or approaching basin size (coverage 80-100%). Zhou et al. (in review) showed that the hydroclimatology of flood events in this region reflects a mixture of flood agents, consisting of thunderstorms and tropical cyclones. The largest fraction of events in the upper tail of flood distributions for basins in this area is associated with organised thunderstorms, which could explain the spatially concentrated nature of storm cores over LSugar Creek subbasins. Table 2 shows the degree of overlap in selected storm events between the 5 (sub-)basins. The table shows that 54% to 69% of events in the largest basin (Lower LSugar) is represented in the flood event catalog for the smaller basins (first row), indicating that these events are likely to have been large-scale events, affecting the entire basin. Overlap between flood-producing events in Upper Briar and Lower Briar is 59%. Lowest overlap occurs for LHope, indicating that a substantial part of flood events in this smaller basin is associated with a different collection of storm events compared to the other basins.

Figure 4 shows scatter plots of fractional coverage versus peak flow. The plots show that there is a tendency for peak flows to increase with fractional coverage and that the top peak flow values are generally associated with 100% basin coverage by the storm core. This confirms results found by Smith et al. (2002) who concluded that the relation between storm scale and basin was an important driver for flood response and Syed et al. (2003) who found that areal coverage of the storm core was better correlated with runoff than area coverage of the entire storm. Our results show that for the urbanised basins in Little Sugar Creek, some of the highest peak flows occur for fractional coverage well below 100%. This could be associated with urbanisation effects changing the upper tail of the peak flow distribution, as was suggested by Zhou et al. (in review), resulting in a different representation of storm events in the highest quantile peak flows.

We analysed relationships between fractional coverage and rainfall intensity to see whether changes in basin-average rainfall are strongly tied to change in fractional coverage by the storm core, associated with the storm core moving into or out of the basin. Spearman rank correlation between 1st order differences in rainfall intensity and rainfall coverage with time ($\Delta R/\Delta t$ versus $\Delta Rcov/\Delta t$) showed high values for all basins, varying from 0.71 for Upper LSugar to 0.84 for Lower Briar. This





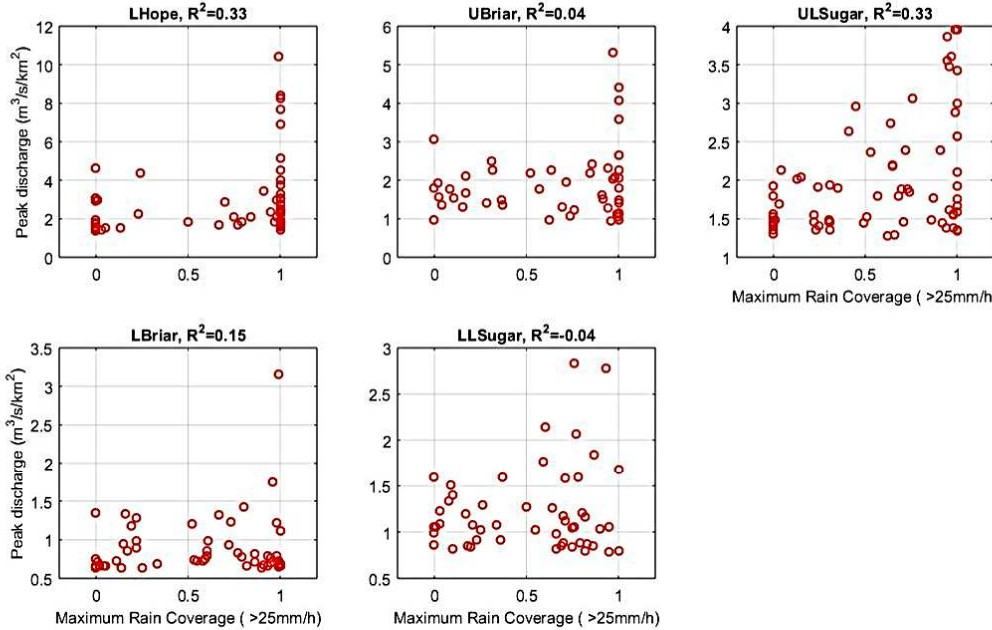

**Figure 4.** Scatter plots of basin fractional coverage by rainfall intensities >25 mm/h versus peak flow, per event, for the five basins in Little Sugar Creek catchment and associated values for Spearman rank correlation coefficients.

confirms that for the selected set of largest flow events in these basins, change in fractional coverage by the storm core is an important driver for change in basin-average rainfall intensity.

### 3.3 Rainfall position and movement relative to flowpath network and effects on hydrological response

An important aim of this study was to investigate how position and movement of rainfall in relation to the flow path network, influences hydrological response. Figure 5 shows time-series of basin-average rainfall, fractional coverage by storm core (>25 mm/h) and RWD and RWD dispersion for two selected events in Lower LSugar basin. The two events (figure 5**a** and 5**b**) represent events from the top-10 highest peak flows in this basin. The third row in the figure illustrates development of RWD as a function of time, the dashed line shows the flow distance for uniform rainfall, 0.53. The figure shows that RWD values vary in a relatively small range around the mean: mean values are 0.41 and 0.40, for a 3-hour time window centered on the rainfall peak. Associated coefficient of variation values are 0.30 and 0.23. This indicates that, on average, rainfall was concentrated slightly closer to the basin outlet compared to uniform rainfall. RWD dispersion shows whether rainfall is distributed





uniformly, unimodally or multimodally with respect to the flowpath network (see also equation 11). Mean RWD dispersion values are 0.83 and 0.93, for a 3-hour window centered on the rainfall peak. Maximum normalised RWD dispersion is 1.04 for the first, 1.39 for the second event. This indicates that on average rainfall was mildly concentrated in space compared to uniform rainfall, the first event being more unimodal and concentrated in space during the peak of the storm and the second

event breaking into a multimodal structure in between the two rainfall peaks. Storm movement relative to the flowpath network can be derived from the time-series of RWD, by analysing gradients in RWD over time. As figure 5 shows, RWD was more or less constant during the period of most intense rainfall for the first event (cf. period with rainfall intensities > 25 mm/h), indicating that storm position relative to the flowpath network changed little during the event. For the second event, RWD decreased from 0.64 to 0.24, the main decrease happening at the same time rainfall intensities decreased. This implies that the

storm moved into the basin at the upstream end of the flowpath network and moved towards the outlet at the end of the event, to about 0.24 of the maximum flow distance (storm centered over the outlet corresponds to flow distance value of zero).

### 3.3.1    Relationship between storm position relative to flowpath network and hydrological response

Figure 6 shows boxplots of RWD values for event-total accumulated rainfall depth (6 **a**) and for mean and gradient of 2-hour

temporally varied RWD (6 **b** and **c**), for the five basins. Results show that variability in RWD tends to be low: 25-75% range smaller than 0.1 for many of the basins. Variability increases with a combination of basin size and shape: largest variability occurs for U LSugar, the most elongated basin (see compactness values table 1). This effect is emphasised for RWD dispersion, where median values are lower and variability is much higher for the larger and elongated basins than for the two small basins ((6 **d**). These results show that rainfall spatial distribution is highly smoothed by the flowpath network and that distribution

of rainfall-weighted flow distances tends to be near uniform for the smallest basins. This suggests that position of the storm relative to the flowpath network is likely to play a role in explaining hydrological response variability mainly in the larger basins. More spatially unimodal events occur in the larger and more elongated basins (6**c**), but variability in position along the flowpath network remains relatively small. This is expected to limit the potential explanatory value of RWD in relation to hydrological response.

Table 3 summarises Spearman rank correlation values between RWD computed for total accumulated rainfall depth per storm event and hydrological response characteristics, peak flow and lag time. Results show that storm-total RWD was not significantly correlated with peak flow. Lag time was significantly and positively correlated with storm-total RWD, for the larger basins (Lower Briar and Lower LSugar) indicating that in these basins, storm events concentrating in the upstream parts of the flowpath network are associated with longer lag times, as illustrated in the scatter plot in (7 **a**).

### 3.3.2    Relationship between storm movement relative to flowpath network and hydrological response

In this section we investigate how the combination of storm position and movement in time influence hydrological response. We analysed correlations with peak flow and lag time for minimum, mean and maximum RWD over a range of time windows. Table 4 summarises correlation values for peak flow and lag time, in relation to rainfall depth, rainfall intensity and mean




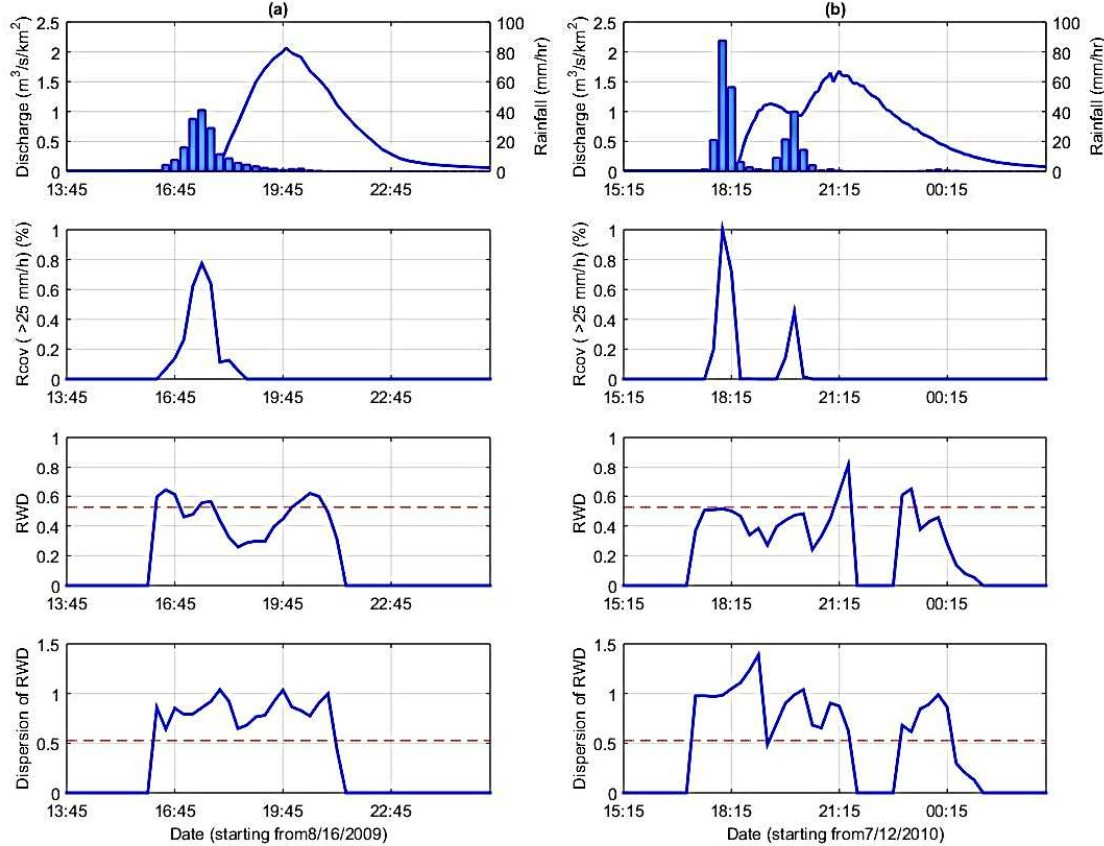

**Figure 5.** Time series of basin-average rainfall, flow, portion of basin covered by high-intensity rainfall (>25 mm/h), normalised rainfall-weighted flow distance (RWD) and RWD dispersion in Lower LSugar, for 3 events that occurred on 16 August 2009 (**a**), 12 July 2010 (**b**) and 28 June 2014 (**c**).

RWD. Highest correlations with peak flow were found for RWD-values associated with a 2 or 2.5 hour time window, except for LHope, where highest correlation was found for mean RWD over a 3-hour time window (Spearman correlation 0.39). This is unexpected, since the mean response time for this basin is only 40 minutes and shortest of all basins. Minimum RWD was significantly correlated with peak flow for the two smallest basins, for a 2.5 hour time window (Spearman correlation 0.31 and 0.30 for LHope and Upper Briar; lower correlations were found for shorter time windows). For these basins, rainfall concentrated in the upstream part of the basins were associated with significantly higher peak flows. Correlations were of the same order of magnitude as those between total rainfall depth or peak rainfall intensity and peak flow (0.30 and 0.40 for LHope, 0.32 and 0.33 for Upper Briar, respectively). For Upper and Lower LSugar, negative correlations were found between RWD and peak flow. Strongest correlations were associated with a 2-hour time window, for mean and maximum RWD (7 **b**): -0.27





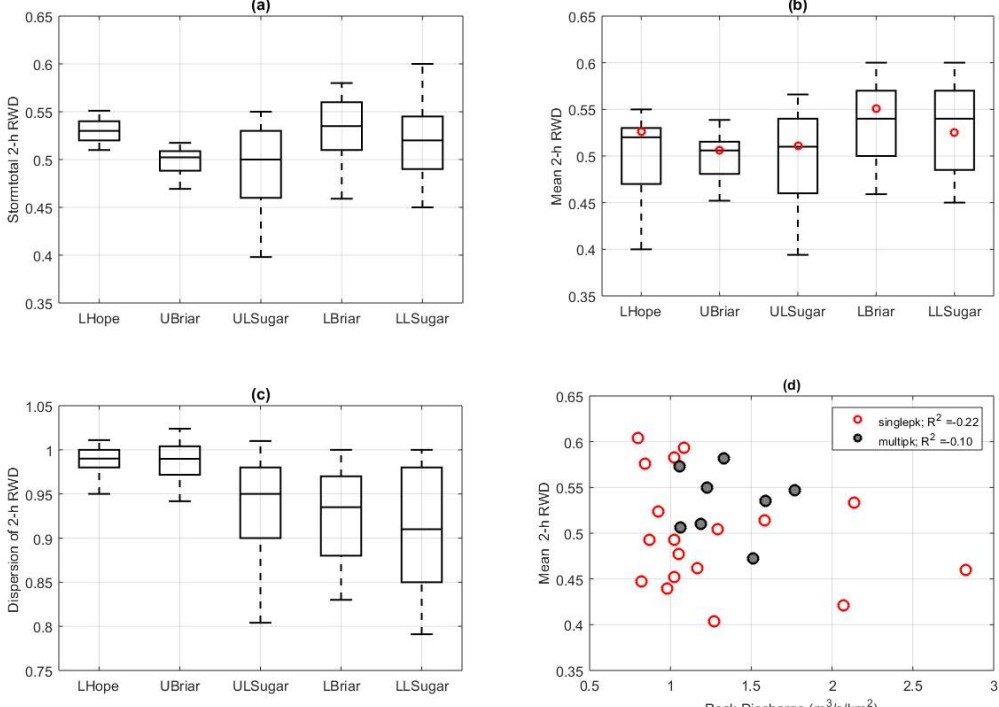

**Figure 6.** Boxplots of RWD values for storm total rainfall (**a**); mean RWD for a 2-hour window and RWD dispersion for a 2-hour window **c**, for all events, for the 5 basins; scatter plot of mean RWD versus peak flow, for Lower LSugar. Red circles in boxplots indicate RWD associated with spatially uniform rainfall

**Table 3.** Spearman rank correlations values and associated p-values for RWD associated with storm event total accumulated rainfall depth versus peak flow (Qp) and lag time (Tlag), for selected time windows centered on the peak flow (1-hour and 2-hour time windows). * indicates significant correlations at the 5% level

| Basin | Qp-RWD(1h) | Qp-RWD(2h) | Tlag-RWD(1h) | Tlag-RWD(2h) |
|---|---|---|---|---|
| LHope | 0.01 | 0.07 | 0.26 | 0.20 |
| UBriar | 0.11 | 0.14 | 0.02 | 0.12 |
| ULSugar | -0.15 | -0.18 | 0.03 | 0.05 |
| LBriar | 0.02 | 0.08 | 0.45* | 0.41* |
| LLSugar | -0.11 | -0.16 | 0.25* | 0.32* |

and -0.32 for Upper LSugar; -0.29 and -0.23 for Lower LSugar, respectively. In these two basins, rainfall concentrated near the outlet was associated with significantly higher peak flows. Correlations were weaker than those between rainfall depth or peak





rainfall intensity and peak flow (0.49 and 0.43 for Upper LSugar; 0.48 and 0.32 for Lower LSugar, respectively). Correlations between rainfall depth and intensity versus mean and maximum RWD were weak, showing that information about the spatial position of rainfall with respect to the flowpath network has added value in explaining peak flow variability. The 2-hour time window for which strongest correlations were found, does not seem to have a relation with lag time, given that Upper LSugar

has a mean lag time of 51 minutes and Lower LSugar of 2.5 hours. A possible explanation for the negative correlation between RWD and peak flow for the Upper and Lower LSugar basins is the spatial distribution of impervious area associated with the urban core of Charlotte. This will be analysed in more detail in section 3.4. No significant correlations were found for Lower Briar, which suggests that spatial rainfall distribution does not influence peak flows, possibly due to a strong smoothing effect of the flowpath network in this relatively large and less urbanised basin.

We separately investigated correlations between rainfall-weighted flow distance and hydrological response for a subset of clear, single-peak events, to exclude more complex correlation patterns associated with multi-peak events. Single peak events tend to show slightly higher correlations compared to multi-peak events, between rainfall properties or rainfall-weighted flow distances and peak flow or lag time (figure 6d). We also investigated whether correlations were different for small-scale storms compared to large-scale storms, by splitting the storm catalog into events with maximum rainfall coverage >25 mm/h above

and below 50%. Correlation values for the two subsets improved for some cases, but improvements were not consistent across different basins. Finally, we investigated correlations for a subset of the storm event catalog, with strong relation between storm movement and rainfall-weighted flow distance, as indicated by strong correlation between, implying that change in rainfall intensity is closely associated with rainfall moving across the basin. The number of events with significant $\Delta R_b/\Delta t$ versus $\Delta D_{Rw}/\Delta t$ correlation varied from 12 for Lower Briar to 22 for Upper LSugar, i.e. 22% to 34% of the storm catalog. Generally,

correlations with peak flow and lag time improved, indicating that storm movement into and out of the basin, leading to changes in basin-average rainfall intensity, significantly contributes to explaining variability in hydrologic response. Investigations for event subsets served as a first exploration of potential multivariate relationships in the datasets. Results showed that explaining variability in hydrological response based on rainfall-weighted flow distance is more straightforward for single peak events than for multi-peak events and that storm movement into and out a basin plays a significant role in explaining variability in

hydrological response.

Table 4 shows that lag time was significantly negatively correlated with gradient in RWD associated with storm movement, for Upper Briar, Upper and Lower LSugar. This implies that storms moving faster towards the basin outlet were associated with slightly shorter longer times. Figure 7d shows that the relationship with flow distance gradient is more subtle: small (near zero) gradients tend to be associated with longer lag times, while fast moving storms tend to be associated with short lag times.

Negative correlation with lag time is explained by negative gradients dominating over positive gradients. Additionally, figure 7c shows that large peak flows tend to occur for gradients near zero, i.e. slow moving, near-stationary storms (relative to the flow path network) or moving storms of larger size than the basin area (especially for smaller basins like LHope).

No significant correlations were found between dispersion of rainfall weighted flow distance and peak flow or lag time, showing that temporal variability in uni- or multimodality of storm events does not have a significant influence on hydrological response.




**Table 4.** Summary of correlations between peak flow (Qpeak), lag time (Tlag) and total basin-average rainfall (Rtot), peak rainfall intensity (Rmax), mean RWD for a 2-hour time window (RWD), minimum RWD for 2.5 hour window (RWDmin) and gradient in RWD for a 2-hour time window (RWDgrad). Significant correlations at the 5% level are indicated in bold

| Basin | Peak flow | | | | Lag Time | | |
|---|---|---|---|---|---|---|---|
| Name | vs Rtot | vs Rmax | vs RWD | vs RWDmin | vs Rtot | vs RWD | vs RWDgrad |
| LHope | **0.30** | **0.40** | **0.31** | **0.31** | **0.39** | 0.18 | -0.08 |
| UBriar | **0.32** | **0.33** | -0.03 | **0.30** | **0.31** | -0.15 | **-0.37** |
| ULSugar | **0.49** | **0.43** | **-0.27** | 0.08 | **0.29** | -0.08 | -0.20 |
| LBriar | **0.53** | **0.38** | 0.06 | 0.10 | **0.56** | **0.25** | -0.09 |
| LLSugar | **0.48** | **0.32** | **-0.29** | 0.11 | **0.43** | 0.05 | **-0.49** |

In this section we analysed influence of position and movement of storms relative to the flowpath network on hydrological response. Results showed that spatial rainfall variability was strongly smoothed by the flowpath network, confirming similar results found by Smith et al. (2005) for a small (14. km$^2$) basin. We found that in small basins rainfall concentrated in the upstream part of the basins was associated with higher peak flows, while in larger basins rainfall concentrated near the outlet was associated with significantly higher peak flows. Correlations were of the same order of magnitude or slightly weaker than those between total rainfall depth or peak rainfall intensity and peak flow. This confirms results found by Smith et al. (2002) who found that for only 1 of 5 storms they analysed, storm position and movement amplified peak flow. While Syed et al. (2003) found that the importance of storm position increased with basin size, this effect was not clearly visible for the basins we investigated in our study. Slow moving, near-stationary storms (relative to the flow path network) were associated with longer lag times in some, but not all basins; near-stationary storms also tend to be associated with higher peak flows. Earlier studies have surmised sensitivity of hydrological response to storm position and movement to be highest when computed over time-windows equal to the basin lag time (Zoccatelli et al. (2011); Nikolopoulos et al. (2014)). In our analyses, we found no relation between time-window for computation of storm position or movement and basin response time.

### 3.4 Spatial distribution of impervious areas, spatial rainfall variability and hydrological response

Spatial distribution of rainfall in relation to distribution of impervious areas in the basins is expected to have an influence on peak flow and lag time, since rainfall on impervious areas generates relatively more runoff and runs off faster compared to pervious areas. The degree of interaction between spatial rainfall variability and spatial imperviousness distribution is likely to depend on two factors: degree of impervious cover in a basin and degree of spatial variation in imperviousness. Figure 8**a** shows the cumulative distribution of basin area as a function of distance along the flowpath network for the five basins in Little Sugar Creek. Figure 8**b** shows the cumulative distribution for impervious areas. Gradients steeper than the 1-to-1 line indicate where basin area, relatively impervious areas are concentrated along the flowpath network.

Imperviousness is most inhomogeneously distributed for LHope, where it is almost entirely concentrated in the upstream part of the basin (above 0.55 normalised distance along the flowpath network). In Upper Briar, impervious areas is more





**Figure 7.** Scatter plots for storm-total RWD (2-hour window) versus lag time (**a**); maximum RWD (2-hour window) versus peak flow (**b**); gradient in RWD (2-hour window) versus peak flow (**c**) and versus lag time (**d**).

concentrated between 0.4 and 0.6 normalised RWD. In Upper LSugar, imperviousness is nearly homogeneously distributed along the flowpath network. In Lower LSugar and Lower Briar impervious areas are slightly more concentrated near and just downstream of the mean flowpath distance.

We analysed the influence of rainfall spatial variability in relation to the distribution of impervious areas based on a binary
5   weighting of RWD by imperviousness, as described in section 2.2.2. We found that variability in RWD per event increased, i.e. mean coefficient of variation in RWD is higher when weighted by imperviousness than for total basin area, for 3 of the 5 basins: LHope, Upper Briar and Lower LSugar. This is illustrated in the scatter plots for RWD and imperviousness-weighted RWD versus peak flow in figure 9. Variability in RWD decreased for Lower Briar. Here, variability in RWD based





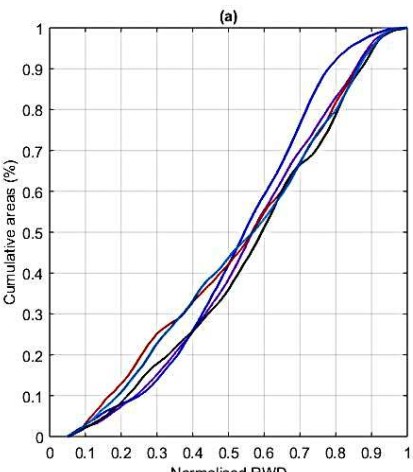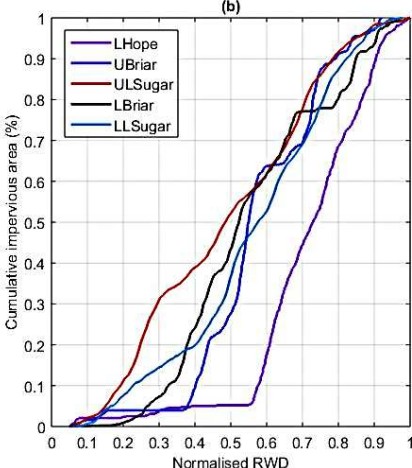

**Figure 8.** Cumulative distribution of catchment area (**a**) and of impervious areas (**a**) as a function of distance along the flow path network, for the five basins in Little Sugar Creek catchment.

on total basin area is higher compared to the other basins with larger area concentrated at the headwaters (see figure 8). Weighting by imperviousness results in a reduction of RWD variability, as impervious areas are situated relatively close to the centre of the flowpath network. For Upper LSugar, weighting by imperviousness has little effect on RWD variability, as could be expected based on the near-homogeneous imperviousness distribution. We analysed influence of imperviousness on hydrological response based on Spearman correlations between imperviousness-weighted RWD, peak flow and lag time. As figure 9 shows, correlations between RWD metrics based on impervious-weighting and peak flow changed little or slightly decreased compared to those based on total basin area. The overall effect was that correlations based on impervious-weighted RWD for both peak flow and lag time were weak and no longer significant at the 5% level. A possible explanation for the limited effect of imperviousness variability on flow peak and lag time is that for the largest basins where spatial rainfall variability is higher, spatial variability in imperviousness is relatively low. Differences in RWD between events increase by imperviousness weighting only for the smallest basin, LHope, while they remain more or less neutral for Upper Briar and Upper LSugar and slightly decrease for Lower Briar and Lower LSugar. This suggests that the effect of increased RWD variability within events by impervious weighting (found for LHope, Upper BRiar and Lower LSugar) is counteracted by reduced variability in RWD between events. Future studies covering a wider range of basin scales and variability in impervious cover will be needed,





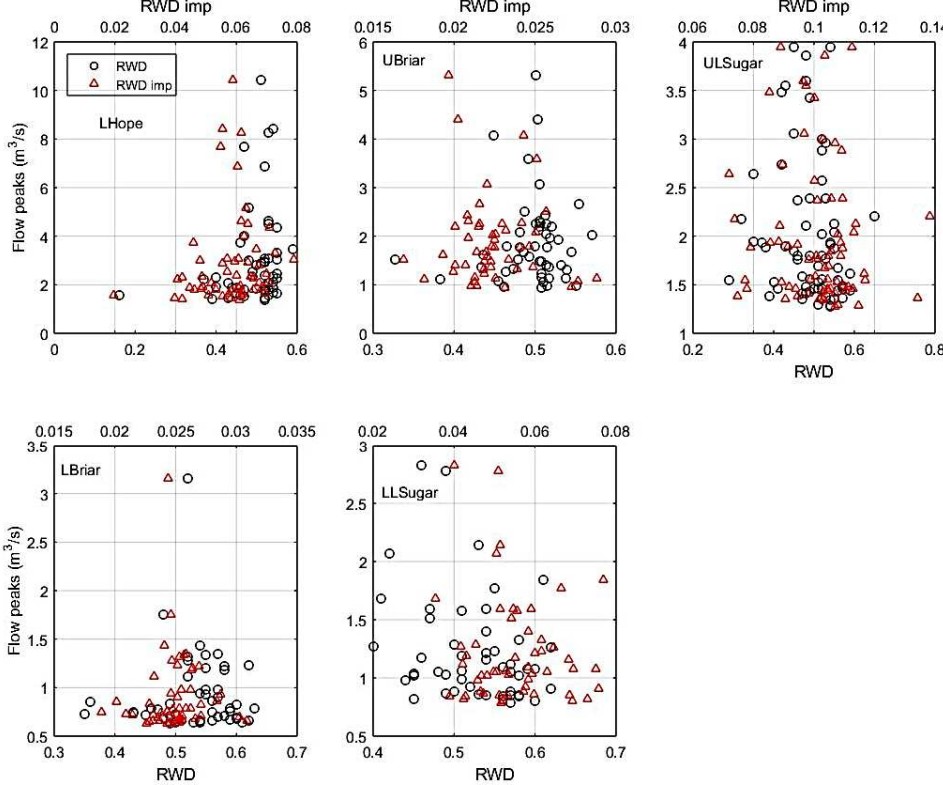

**Figure 9.** Scatter plots of 2h-mean RWD versus peak flow, for RWD based on all areas (lower x-axis) and for RWD weighted by imperviousness (upper x-axis), for the five basins in LSugar Creek catchment.

to investigate to what extent this conclusion holds for other urbanised basins and how the lack of sensitivity to impervious cover can be explained. Apart from impervious cover, the effect of spatial distribution of urban soils with relatively lower permeability than natural soils, can be analysed using the same approach. This will provide better insights into characteristic imperviousness cover and variability scales that determine sensitivity of hydrological response to spatial rainfall variability.

## 4   Summary and conclusions

The objective of this study is to provide insights into how rainfall spatial and temporal variability interact with catchment scale and flowpath network structure in generating hydrological response in urbanised basins, based on extensive observational datasets. The study comprised analysis of a catalog of the largest 279 flood events extracted from 15 years of rainfall and flow data over 5 nested basins of varying size and degree of urban development. We analysed rainfall coverage over the basin and over impervious areas in the basin to analyse spatial variability effects on peak flow and lag time. We used the concept of





rainfall-weighted flow distance introduced by Smith et al. (2002) to analyse how storm position and movement relative to the flowpath network influenced hydrological response. The following conclusions were drawn from the analyses:

1. Catchment scale determines the type of storm events that produce largest basin peak flows: storm events for the catalog of largest peak flows in the small, 7 km$^2$ basin, show only 39-54% overlap with those for the other basins. Largest overlap in storm events, 69%, is found for the two larger basins, 48.5 and 111.1 km$^2$ in size. This confirms results reported by Smith et al. (2013b) and Zhou et al. (in review), who also found markedly different rainfall climatologies for flood-producing storms in basins of different size.

2. Catchment scale determines the degree of variability in peak flows and peak rainfall intensities for the catalog of largest flood events. Coefficient of variation in peak flows varies from 0.46 for the largest to 0.65 for the smallest basin. Lowest flow variability is found for the most urbanised basin (size 31.5 km$^2$), which suggests a smoothing effect of imperviousness on flow variability. Similar results were found by other authors and were attributed to the effect of constraints in the drainage network (Smith and Smith (2015); ten Veldhuis and Schleiss (accepted for publication).

3. Storm scales, measured by maximum coverage of a basin by rainfall intensities above 25 mm/h, representing the most intense storm core, vary strongly with basin scale: for the smallest, 7 km$^2$ basin, intense storm core exceeds basin scale for 43% of the storms, while 30% of the storms cover less than half of the basin. For the largest basin, storm core exceeds basin scale for only 2% of the storms and 44% of events cover less than half the basin area. Empirical histograms of rainfall coverage for intensities above 25 mm/h show that for the smaller basins, up to 31.5 km$^2$, storm events largely fall into two groups: large-scale events, with intense storm core exceeding basin scale and small-scale events, with storm core covering less than 20% of the basin.

4. Dynamics of rainfall coverage are an important driver for basin-average rainfall variability; Spearman correlation is around and above 0.8 for the five basins. This suggests that storm movement over the basin drives increase and decrease in basin-average rainfall intensity more strongly than development of storm cells during storm passage over the basin. Maximum rainfall coverage (>25 mm/h) is significantly and positively correlated with peak flow for two of the five basins: the smallest and the most impervious basin (Spearman correlation 0.33 for both). No significant correlation was found for the other basins. This contrasts with results found by Smith et al. (2002) and Syed et al. (2003), who concluded that fractional coverage by the storm core plays a more important role for larger basins.

5. The combination of spatial rainfall structure and flowpath network (expressed in terms of rainfall-weighted flow distance) plays a smaller role in explaining variability in hydrological response compared to rainfall volume and peak intensity. This could be explained by spatial rainfall variability having a relatively small contribution to flow variability compared to climatological rainfall variability, as shown by Peleg et al. (2017). Another explanation is that rainfall spatial variability is strongly smoothed by the flowpath network, as was also shown in earlier studies for a more limited range of observations (Smith et al., 2005).





6. The role of storm movement relative to the flow path network is investigated based on temporal gradients in rainfall-weighted flow distance. Movement of storms upstream or downstream along the main axis of the flowpath network have no significant influence on peak flows. Slow moving, (near) stationary storms relative to the flowpath network tend to be associated with higher peak flows. Additionally, slow moving storms are generally associated with longer lag times.

7. Impact of spatial variability in urbanisation on hydrological response is investigated based on rainfall-weighted flow distance over impervious areas. We find that urbanisation plays a minor role in explaining variability in peak flow and lag time in the five basins in Little Sugar Creek. A possible explanation is that for the largest basins, where spatial rainfall variability is higher, variability in imperviousness is relatively low. Differences in RWD between events increase by imperviousness weighting only for the smallest basin, LHope, while they remain more or less neutral for Upper Briar and Upper LSugar and slightly decrease for Lower Briar and Lower LSugar. The overall effect is neutral or a weakening of the relationship with hydrological response.

Results of this study based on 279 flood events for a range of basin sizes, clearly show that the relation between rainfall and basin scales is an important driver for generating largest peak flows. Rainfall spatial structure and storm movement seem to play a less important role, being strongly smoothed by the flowpath network. Additional analyses for a larger number of basins are needed to further look into the role of storm position and movement in generating hydrological response. Additionally, the influence of spatial variability in impervious cover on peak flows and lag time needs further investigation to better understand the interplay between spatial distribution of rainfall and urbanisation. The role of other spatially variable catchment characteristics like topography and (urban) soil properties have not been considered in this study, while previous studies have pointed at the role of soil moisture in generation of peak flows that remains poorly understood. The importance of variability in topography, soil moisture and urbanisation in relation to spatial rainfall variability and climatological variability remain important topics for future research. Future work will focus on analyses for a larger number of basins and a larger set of storms, including smaller, more concentrated storms relative to the catchment scale, to investigate the role spatial rainfall variability compared to climatological rainfall variability in explaining hydrological response.

*Acknowledgements.* Flow data used in this study are available at http://waterdata.usgs.gov/nc/nwis. The radar rainfall data used in this study can be made available upon request. We are grateful to M.L. Baeck for making available the NEXRAD radar rainfall datasets used in this study. We would like to thank J. Signel for organising the USGS flow datasets for the hydrological basins in Charlotte. The first author would like to thank NWO Aspasia and Delft University of Technology for the grant that supported this research collaboration. The authors would like to acknowledge support for this research from the National Science Foundation (CBET-1444758 and AGS-1522492 )



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
