# Peer review of "The role of storm scale, position and movement in controlling urban flood response"

_Hydrology and Earth System Sciences, 2017_

## Referee Comment (RC1) · S. Thorndahl (Referee) · 5 Jul 2017

The paper presents interesting data-driven analyses of rainfall-runoff processes for flood events using a unique dataset of stream gauges in combination with radar rainfall data.

The paper is well written and easily understood and my comments below are primarily suggestions to further analyses rather than criticism of the conducted work.

1. Page 6 line 27 – page 7 line 9: Please provide information on the 15 min. radardata. Is this an average over 15 min or does it represent a "snaphot-value" in 15 min window? If it is an average over the 15 min I find it difficult to justify the resampling to 30 m resolution since the rainfall can have moved significantly during the 15 min. The must

thus be a quite large uncertainty related to the time lag between rainfall and flow and the RWD. We did some studies on advection interpolation of "snapshot" radar data in order to increase the temporal resolution (Nielsen et al., 2014; Thorndahl et al., 2014) which gave much better rainfall estimates (doing mean field bias adjustment) than with the data with a lower temporal resolution. In this case we both resampled in time and space. Maybe this could also have been relevant here in order to reduce the aforementioned uncertainty.

2. I think it could be relevant to address the range of return periods of the analyzed events both in terms of rainfall over specific durations (and areas or fractional coverages) as well as return periods of the flood peaks.

3. The definition of flood is somewhat unclear. I guess that many of the events does not actually produce a flood (in the definition of inundation), but more high flows. Maybe it could be relevant to show an hydrograph example related to the definition in page 8 line 16-18.

4. One thing which also could be relevant to consider is the time between rainfall events or the time since the last rainfall event and how that affects the flood peaks. I could imagine that higher saturated soils (as a result of recent rainfall) would correlate well to the flow peaks

5. The use of the empirical 25 mm/h threshold to represent high intensity rainfall could be reasoned better. Would it make any difference if this threshold was lower or higher.

Specific comments

Page2 line 10. Here it could be relevant also to cite Thorndahl et al. (2017)

Equation 2. The use of T is somewhat misleading since it is used twice in the equation.

Figure 2. I could be relevant to provide the number of events in each basin in the figure.

References

Nielsen, J.E., Thorndahl, S., Rasmussen, M.R., 2014. Improving weather radar precipitation estimates by combining two types of radars. Atmospheric Research 139, 36–45. doi:10.1016/j.atmosres.2013.12.013

Thorndahl, S., Einfalt, T., Willems, P., Nielsen, J.E., ten Veldhuis, M.-C., Arnbjerg-Nielsen, K., Rasmussen, M.R., Molnar, P., 2017. Weather radar rainfall data in urban hydrology. Hydrology and Earth System Sciences 21, 1359–1380. doi:10.5194/hess-21-1359-2017

Thorndahl, S., Nielsen, J.E., Rasmussen, M.R., 2014. Bias adjustment and advection interpolation of long-term high resolution radar rainfall series. Journal of Hydrology 508, 214–226. doi:10.1016/j.jhydrol.2013.10.056

---

## Referee Comment (RC2) · Anonymous Referee #2 · 17 Jul 2017

The authors carried out data-driven assessment of the relationship between rainfall variability and streamflow response at catchment outlets for 5 urban catchments in the Charlotte, NC, area. This area has a relatively dense network of stream gauges and high-quality historical data to allow such a study. Though spatial variability of rainfall and land cover is reflected via fractional coverage, radar rainfall estimates and impervious cover, the study is largely about catchment scale response. Though mentioned in the title, this study has little to do with storm dynamics. The authors describe various analyses, largely statistical in nature, carried out using the above data along with the NEXRAD-based rainfall estimates. They arrive at 7 specific conclusions.

I have a number of major issues, including a few pertaining specifically to methodology, as elaborated below.

[Figure]

Major comments

1. Methodology

In my view, the authors' data-driven, largely statistical, analysis could benefit greatly from drawing from the vast literature on modeling studies as well as from applying simple modeling approaches. While I appreciate the motivation for the data-driven approach, I find that the authors are left to connect the dots based almost exclusively on somewhat tenuous observations from noisy data points and a very small number of publications by the same group. My visual examination of the figures in the manuscript suggests that, while various statistical analyses and testing were carried out, the results are overall less than convincing. Calculating correlation to highly nonlinear data, for example, is not appropriate.

In my opinion, deriving empirical unit hydrograph for each catchment at least for a sizable number of single-pulsed events will shed light to the results very significantly. As far as I can tell, the authors have the data to do this. Solving this inverse problem is tricky but doable, given that the authors have high-resolution rainfall and streamflow data. Such analysis would also be entirely in line with the data-driven approach.

1.1 Use of radar data (or lack thereof)

In my view, the authors over-rely on the RWD analysis which is basically a proxy for excess rainfall (or runoff depth)-weighted travel time to the outlet. Because it does not account for spatially-varying velocities, attenuation effects, storage effects, nonlinear effects and integration effects, I do not think it is very amenable to quantitative analysis other than perhaps using as an index to infer the general location of the precipitation core relative to the outlet. If that is the case, I strongly think that the authors are better off examining the radar rainfall data directly. They will show with great certainty where the heavy rainfall was and in which direction the storm was moving, etc. Similarly, I find the exposition on storm vs. catchment scale to be a rather roundabout way to deal with the issue. It would be quite straightforward to characterize the size of the heavy

rain cores directly from the radar rainfall data.

1.2 Stormwater infrastructure

The authors acknowledge its existence, including dams, but it is completely unclear what they are and what impact they may have. Because the size of the storms that the authors are dealing with is small (the largest several events per year), one would expect potentially significant impact by the storm drain network. The impact by the dams and other detention structures would potentially be greater. Little of this, however, is explained or justified.

1.3 Flowpath analysis

It is completely unclear how this was done. Is this meant to capture channel flows only or both channel and hillslope flows? In their lag time analysis, how did they account to spatially varying roughness/velocity? The nature of this analysis has large implications in interpretation of the results.

2 General lack of clarity and specificity

I find the manuscript a very difficult and frustrating read due to loose notations and very liberal use of certain expressions. I illustrate this using a couple of examples below.

Hydrologic response – I am not sure exactly what the authors mean by this expression which is used numerous times throughout the manuscript. In this work, the authors deal with streamflow response at the catchment and subcatchment outlets only. Urban flooding is a concern not only along the main channels, for whose response the outlet flow is a reasonable descriptor, but also in all upstream areas. I was led to believe by the title that this study deals with the role of spatiotemporal variability of rainfall on urban flooding across scale but it is largely about catchment- and subcatchment-wide response to rainfall.

Variability - The authors introduce many different types of variability in the manuscript: spatial variability, temporal variability, catchment variability, flow variability, peak flow

variability, lag time variability, variability in runoff ratio expressed in terms of CV, climatological variability and possibly more. Many of these expressions are, however, rather loosely defined or undefined. For example, by "climatological variability", I believe the authors mean event-to-event variability. Also, fractional coverage is part of spatial variability of rainfall. If the authors mean inner variability, i.e., variability of positive rainfall by "variability of rainfall", they should indicate as such. If CV is used to measure variability, the authors should clearly state of what quantity, if not the complete mathematical expression. Again, the numerous loose descriptions, definitions and notations (see below) make reading this manuscript rather frustrating in that one has to guess at what the authors may actually mean.

**3 Inconsistent and missing notations**

There are many places where the notations are missing, inconsistent, if not incorrect, or confusing. For example, on page 9, $r$ and $r(t,x)$ are never defined. If they mean the same, this is an abuse of notation as the former is a variable and the latter is a function. Also, the usual notation would be $r(x,t)$, not $r(t,x)$. Neither is $DRw(t)$ defined. I do not see how $D(t)$ is a random variable that takes values from 0 to 1. According to Eqs.(8) and (9), if there is excess runoff at time $t$, $D(t)$ should be zero (assuming $r(t,x)$ denotes rainfall at time $t$ and flow path $x$). And yet, in Fig 5, RWD seems to be positive even when $r(t,x)$ is zero.

**4 Significance**

There are 7 specific conclusions the authors draw from this work which are stated in the Summary and Conclusions Section as well as in the abstract. In my view, most of them are already well known and established. I suspect that most practicing hydrologists and water resources engineers, particularly in urban areas, would find them largely a restatement of what they already know and practice.

For the last "unexpected" conclusion, the authors state "We find that urbanisation plays a minor role in explaining variability in peak flow and lag time in the five basins in Little

Sugar Creek." It is not completely clear what is meant by "variability of peak flow and lag time" but, assuming the authors meant event-to-event variability, the above is explained by the following two observations. The first is that these are small catchments (∼111.1 km2) and hence, when there is heavy rainfall, it is very likely rain over most or all of the catchment area. This greatly reduces the likelihood of impervious areas amplifying event-to-event variability in runoff generation as they will almost always generate runoff. The second is that, unlike pervious areas, impervious areas will run off essentially all rainfall. As such, there is little event-to-event variability to be expected over impervious areas in small catchments.

---

## Referee Comment (RC3) · Anonymous Referee #3 · 28 Sep 2017

This paper presents a thorough and well-presented empirical analysis of storm rainfall and runoff across a number of highly urban basins. It is perhaps overly ambitious in brining so many facets together in one paper, leading to some difficulty for the reader to seperate each of the analyses undertaken, but this is balanced by high quality analysis on a large number of high resolution flood events across 5 basins. The paper has 7 substantial conclusions, and each of them is based on a sound analysis of robust data. The language and presentation is overall good, and the paper is well presented.

Specific comments

The last two sentences of the abstract are unclear and unjustified – they can be improved easily.

The role of soil moisture has not been considered in the paper – can the authors comment and justify on why this has not been considered in their analyses.

Figure 2 and the data– are the event data normal and if not have they been normalized before statistical comparison between events. Also – for rainfall, are the rainfall events in fact not independent – and does this not affect the validity of any comparison between catchments if indeed what is being compared is essentially the same rainfall events that pass over them all? Which sites are significantly different in the plot?

Figure 5 – what is the z axis scale on line 2 – 0-1%? I assume it means 0-100%. Also one of the plots then exceeds 100% in the graphic.

The general layout is difficult to follow as tables are referenced well before they are placed in the document – which can make the paper hard to follow – can this be improved in the final manuscript (e.g. Table 4).

Im confused with Tables 3 and 4 and how they are used in the conclusions – please address the following points. In table 3 you state associated p values are set out, but I see no actual reported p values, only asterix to indicate a p value that is significant, here at 5%. Next in table 4 the significant correlations are in bold, rather than asterix being used. In both it seems Spearman rank correlation and significance - see Table 3 where LLsugar has a 0.25* for Tlag-RWD(ih), while LHope has 0.26 - are not related. First in conclusion 4 its stated that dynamics of rainfall coverage are important drivers of rainfall variability – with spearman ranks values exceeding 0.8 for the five basins – from where is this data taken or reported in the paper – what table reports this? Next in conclusion 4 you note maximum rainfall coverage (storm core?) is significantly and positively correlated with peak flow for two of the five basins (smallest and largest), with values of 0.33, and not significant correlation in the others. Again I cannot seem to link this reporting to the results in text or table. The only 0.33 reported is for UBriar in table 4 and also referred to in the text.

I feel conclusion 7 is interesting and warrants further discussion or possible explanation

– as urbanisation more than doubles in some catchments and the general consensus is more urbanisation equals more runoff and higher peak flows. This should also include some caveat regarding the fact storm water infrastructure was not included.

---

## Author Comment (AC1)

Interactive comments on "The role of storm dynamics and scale in controlling urban flood response"

Reviewer 1:
The paper presents interesting data-driven analyses of rainfall-runoff processes for flood events using a unique dataset of stream gauges in combination with radar rainfall data.
The paper is well written and easily understood and my comments below are primarily suggestions to further analyses rather than criticism of the conducted work.
AR: we thank the reviewer for his positive comments and valuable suggestions to which we respond below.

1. Page 6 line 27 – page 7 line 9: Please provide information on the 15 min. radardata.
Is this an average over 15 min or does it represent a "snaphot-value" in 15 min window?
If it is an average over the 15 min I find it difficult to justify the resampling to 30 m resolution since the rainfall can have moved significantly during the 15 min. The must thus be a quite large uncertainty related to the time lag between rainfall and flow and
the RWD. We did some studies on advection interpolation of "snapshot" radar data in order to increase the temporal resolution (Nielsen et al., 2014; Thorndahl et al., 2014) which gave much better rainfall estimates (doing mean field bias adjustment) than with the data with a lower temporal resolution. In this case we both resampled in time and space. Maybe this could also have been relevant here in order to reduce the aforementioned uncertainty.
AR: the 15 minute radar rainfall estimates are "snapshot-values", i.e. the radar beam passes over the area once every 15 minutes. We will clarify this in the data description in section 2.1.
The reason we interpolate to 30 meters is in order to estimate variability in travel distances associated with topography and imperviousness, within the radar pixel. We are aware that the snapshots may not be representative of the entire 15 minute interval, especially for fast moving storms. However, we believe that it sufficiently captures the information for the purpose of our study, i.e. minimum, mean and maximum distance of storm relative to the outlet and movement of storms relative to the flowpath network.

2. I think it could be relevant to address the range of return periods of the analyzed events both in terms of rainfall over specific durations (and areas or fractional coverages) as well as return periods of the flood peaks.
AR: Thanks for this suggestion. We will report return periods for rainfall based on rainfall frequency distributions provided by NOAA for this area*. NOAA provides point rainfall frequency estimates; the closest we can compare to is maximum rainfall intensity values per radar pixel (1x1km2).
Maximum values for 15-min, 1x1 km2 rainfall intensities per event varied from 8.8. to 132 mm/h, associated with return intervals of less than 1 year up to 25 years.
* (https://hdsc.nws.noaa.gov/hdsc/pfds/pfds_map_cont.html?bkmrk=nc)

Based on Villarini et al (2009), who reported flood frequency distributions for LowerLittle Sugar Creek and for LHope Creek, flow peaks for our event catalog (max flow peaks per basin resp. 3.4 and 10.4 m3/s/km2) were associated with 100-year return periods in resp. 1990 and 1992, decreasing to 8 resp. 20 years in 2007, according to the Generalised Additive Model they fitted to annual flood peaks in these 2 basins.

3. The definition of flood is somewhat unclear. I guess that many of the events does not actually produce a flood (in the definition of inundation), but more high flows. Maybe it could be relevant to show an hydrograph example related to the definition in page 8

line 16-18.
AR: thanks for the suggestion. The term "flood response" is indeed used to refer to hydrological response associated with high flow events, in this case the top 5 flow peaks per year, on average. In the catchments that we investigated, it is hard to distinguish between bank-full flow and inundating flows, as channels and natural floodplains were heavily modified as a consequence of urbanisation. As a result, what used to be considered "bank-full" flow in a natural channel could be considered flooding (of private properties, gardens) in the urbanised context (Turner-Gillespie et al., 2003).

Reference:
Turner-Gillespie, D. F., Smith, J. A., & Bates, P. D. (2003). Attenuating reaches and the regional flood response of an urbanizing drainage basin. Advances in Water Resources, 26(6), 673-684.

4. One thing which also could be relevant to consider is the time between rainfall events or the time since the last rainfall event and how that affects the flood peaks. I could imagine that higher saturated soils (as a result of recent rainfall) would correlate well to the flow peaks
AR: This is indeed a relevant point that has been investigated in previous publications, incl a recent paper by Zhou et al. (in press, WRR). They did not find a clear relationship between watershed wetness (represented by antecedent rainfall and streamflow) and flow peaks. This is why we have chosen not to include this as a potential explanatory variable in our analysis.

5. The use of the empirical 25 mm/h threshold to represent high intensity rainfall could be reasoned better. Would it make any difference if this threshold was lower or higher.
AR: We chose the 25 mm/h threshold as it corresponds with the 1 inch threshold that is used by the flood hazard community, specifically the National Weather Service, as an index for potential flash flooding. It has also been used previously in the literature to investigate the influence of storm core versus overall rainfall (e.g. Syed et al., 2003).

Specific comments
Page2 line 10. Here it could be relevant also to cite Thorndahl et al. (2017)
AR: thanks for the suggestion, we will add the citation
Equation 2. The use of T is somewhat misleading since it is used twice in the equation.
AR: thanks for pointing this out, we will correct the equation to avoid confusion
Figure 2. I could be relevant to provide the number of events in each basin in the figure.
AR: the number of events per basin are provided in table 3. We prefer not to add the number of events in the figure in order not to crowd the plots. We can add the information in the figure caption.

References
Nielsen, J.E., Thorndahl, S., Rasmussen, M.R., 2014. Improving weather radar precipitation estimates by combining two types of radars. Atmospheric Research 139, 36–45. doi:10.1016/j.atmosres.2013.12.013
Thorndahl, S., Einfalt, T., Willems, P., Nielsen, J.E., ten Veldhuis, M.-C., Arnbjerg-Nielsen, K., Rasmussen, M.R., Molnar, P., 2017. Weather radar rainfall data in urban hydrology. Hydrology and Earth System Sciences 21, 1359–1380. doi:10.5194/hess-21-1359-2017
Thorndahl, S., Nielsen, J.E., Rasmussen, M.R., 2014. Bias adjustment and advection interpolation of long-term high resolution radar rainfall series. Journal of Hydrology 508, 214–226. doi:10.1016/j.jhydrol.2013.10.056

---

## Author Response (AR1)

Interactive comments on "The role of storm dynamics and scale in controlling urban flood response"

Reviewer 1:
The paper presents interesting data-driven analyses of rainfall-runoff processes for flood events using a unique dataset of stream gauges in combination with radar rainfall data.
The paper is well written and easily understood and my comments below are primarily suggestions to further analyses rather than criticism of the conducted work.
AR: we thank the reviewer for his positive comments and valuable suggestions to which we respond below.

1. Page 6 line 27 – page 7 line 9: Please provide information on the 15 min. radar data. Is this an average over 15 min or does it represent a "snaphot-value" in 15 min window? If it is an average over the 15 min I find it difficult to justify the resampling to 30 m resolution since the rainfall can have moved significantly during the 15 min. The must thus be a quite large uncertainty related to the time lag between rainfall and flow and the RWD. We did some studies on advection interpolation of "snapshot" radar data in order to increase the temporal resolution (Nielsen et al., 2014; Thorndahl et al., 2014) which gave much better rainfall estimates (doing mean field bias adjustment) than with the data with a lower temporal resolution. In this case we both resampled in time and space. Maybe this could also have been relevant here in order to reduce the aforementioned uncertainty.
AR: the 15 minute radar rainfall estimates are based on "snapshot-values" constructed on volume scan times, which have a time scale of 5-6 minutes. We have clarified this in the data description in section 2.1.
The reason we interpolate to 30 meters is in order to estimate variability in travel distances associated with topography and imperviousness, within the radar pixel. We are aware that the snapshots may not be representative of the entire 15 minute interval, especially for fast moving storms. However, we believe that it sufficiently captures the information for the purpose of our study, i.e. minimum, mean and maximum distance of storm relative to the outlet and movement of storms relative to the flowpath network.
The following explanation was added (P7, L12-15): "While 15-minute estimates derived from 5-minute radar sampling may smooth some of the rainfall variability, especially for fast moving storms, they sufficiently capture the rainfall information relevant for this study, i.e. minimum, mean and maximum distance of storms relative to the outlet and movement of storms relative to the flowpath network."

2. I think it could be relevant to address the range of return periods of the analysed events both in terms of rainfall over specific durations (and areas or fractional coverages) as well as return periods of the flood peaks.
AR: Thanks for this suggestion. We will report return periods for rainfall based on rainfall frequency distributions provided by NOAA for this area*. NOAA provides point rainfall frequency estimates; the closest we can compare to is maximum rainfall intensity values per radar pixel (1x1km2).
Maximum values for 15-min, 1x1 km2 rainfall intensities per event varied from 8.8. to 132 mm/h, associated with return intervals of less than 1 year up to 25 years.
* (https://hdsc.nws.noaa.gov/hdsc/pfds/pfds_map_cont.html?bkmrk=nc)
Based on Villarini et al (2009), who reported flood frequency distributions for Lower LSugar Creek and for LHope Creek, flow peaks for our event catalog (max flow peaks per basin resp. 3.4 and 10.4 m3/s/km2) were associated with 100-year return periods in resp. 1990 and 1992, decreasing to 8 resp. 20 years in 2007, according to the Generalised Additive Model they fitted to annual flood peaks in these 2 basins.
The following explanation was added in section 3.1 (p13, L7-13): "Flow peaks for our event catalog (max flow peaks per basin resp. 3.4 and 10.4 m3/s/km2) were associated with 100-year return periods in resp. 1990 and 1992, decreasing to 8 resp. 20 years in 2007, following Villarini et al. (2009), who reported flood frequency distributions for Lower LSugar Creek and for LHope Creek, based on a

Generalised Additive Model fitted to annual flood peaks in these 2 basins. For rainfall, we compared return intervals of maximum 15-minute rainfall intensities (over 1x1 km2 with point rainfall frequency estimates provided by NOAA; no frequency estimates were available at 1x1 km2 scale. Maximum values per event varied from 8.8 to 132 mm/h, associated with return intervals of less than 1 year up to 25 years at the point scale."

3. The definition of flood is somewhat unclear. I guess that many of the events does not actually produce a flood (in the definition of inundation), but more high flows. Maybe it could be relevant to show a hydrograph example related to the definition in page 8
line 16-18.
AR: thanks for the suggestion, the following text was added for clarification (P4, L12-16): " The term "flood response" is used to refer to hydrological response associated with these high flow events, at the (sub)catchment scale. In the catchments we investigated, it is hard to distinguish between bank-full flow and inundating flows, since channels and natural floodplains were heavily modified as a consequence of urbanisation. As a result, what used to be considered "bank-full" flow in a natural channel could be considered flooding (of private properties, gardens) in the urbanised context (Turner-Gillespie et al., 2003)."

4. One thing which also could be relevant to consider is the time between rainfall events or the time since the last rainfall event and how that affects the flood peaks. I could imagine that higher saturated soils (as a result of recent rainfall) would correlate well to the flow peaks
AR: This is indeed a relevant point that has been investigated in previous publications, incl. a recent paper by Zhou et al. (2017). They did not find a clear relationship between watershed wetness (represented by antecedent rainfall and streamflow) and flow peaks. This is why we have chosen not to include this as a potential explanatory variable in our analysis.
The following text was added in section 1 (p. 4): "Our study focuses on spatial storm characteristics and does not look at effects of time between rainfall events or the time since the last rainfall event and how that affects the flood peaks. In a recent study by Zhou et al. (2017) the effect of watershed wetness was investigated for the Charlotte region; they did not find a significant influence of antecedent rainfall on flood response."

5. The use of the empirical 25 mm/h threshold to represent high intensity rainfall could be reasoned better. Would it make any difference if this threshold was lower or higher.
AR: We chose the 25 mm/h threshold as it corresponds with the 1 inch threshold that is used by the flood hazard community, specifically the National Weather Service, as an index for potential flash flooding. It has also been used previously in the literature to investigate the influence of storm core versus overall rainfall (e.g. Syed et al., 2003).
Explanation has been added in section 2.2.2 (p9, L27-30).

Specific comments
Page2 line 10. Here it could be relevant also to cite Thorndahl et al. (2017)
AR: thanks for the suggestion, we have added this citation.
Equation 2. The use of T is somewhat misleading since it is used twice in the equation.
AR: thanks for pointing this out, we have corrected the equation to avoid confusion
Figure 2. I could be relevant to provide the number of events in each basin in the figure.
AR: the number of events per basin are provided in table 1. We prefer not to add the number of events in the figure in order not to crowd the plots. We have added a reference to table 1 in the figure caption.

References
Nielsen, J.E., Thorndahl, S., Rasmussen, M.R., 2014. Improving weather radar precipitation

estimates by combining two types of radars. Atmospheric Research 139, 36–45. doi:10.1016/j.atmosres.2013.12.013

Thorndahl, S., Einfalt, T., Willems, P., Nielsen, J.E., ten Veldhuis, M.-C., Arnbjerg-Nielsen, K., Rasmussen, M.R., Molnar, P., 2017. Weather radar rainfall data in urban hydrology. Hydrology and Earth System Sciences 21, 1359–1380. doi:10.5194/hess-21-1359-2017

Thorndahl, S., Nielsen, J.E., Rasmussen, M.R., 2014. Bias adjustment and advection interpolation of long-term high resolution radar rainfall series. Journal of Hydrology 508, 214–226. doi:10.1016/j.jhydrol.2013.10.056

Interactive comments on "The role of storm dynamics and scale in controlling urban flood response"

Reviewer 2:
The authors carried out data-driven assessment of the relationship between rainfall variability and streamflow response at catchment outlets for 5 urban catchments in the Charlotte, NC, area. This area has a relatively dense network of stream gauges and high-quality historical data to allow such a study. Though spatial variability of rainfall and land cover is reflected via fractional coverage, radar rainfall estimates and impervious cover, the study is largely about catchment scale response. Though mentioned in the title, this study has little to do with storm dynamics. The authors describe various analyses, largely statistical in nature, carried out using the above data along with the NEXRAD-based rainfall estimates. They arrive at 7 specific conclusions.
I have a number of major issues, including a few pertaining specifically to methodology, as elaborated below.
AR: We infer from the reviewer's comments that he/she expected a very different manuscript when accepting to review. We understand the disappointment and see that it has led to misunderstanding in the interpretation of the manuscript. We will adjust our phrasing for those instances that seem to have caused the confusion: "storm dynamics and scale" in the title will be replaced by "storm position, movement and scale. Similarly, "rainfall spatial distribution" in the abstract will be replaced by "storm position, movement and scale" to be more explicit about what spatial aspects of rainfall were studied and what we mean by storm dynamics.
Major comments
1. Methodology
In my view, the authors' data-driven, largely statistical, analysis could benefit greatly from drawing from the vast literature on modeling studies as well as from applying simple modeling approaches. While I appreciate the motivation for the data-driven approach, I find that the authors are left to connect the dots based almost exclusively on somewhat tenuous observations from noisy data points and a very small number of publications by the same group.
AR: We thank the reviewer for his appreciation of the data-driven approach. He/she is right in that deriving conclusions from field observations is challenging, given the complex nature of the processes involved. The opportunity to study urban flood response based on such long records of combined radar-rainfall and flow observations is unprecedented. We want to emphasise that the datasets are of high quality, hence, what we see represented in the data is not noisiness, but complexity of the underlying processes. Statistical analyses allow us to identify critical parameters for describing flood response, without the need of making any pre-assumptions as in an empirical modelling approach.

My visual examination of the figures in the manuscript suggests that, while various statistical analyses and testing were carried out, the results are overall less than convincing. Calculating correlation to highly nonlinear data, for example, is not appropriate.
AR: We are fully aware of the non-linearity of the processes we're studying; to deal with this, we have used Spearman rank correlations (not Pearson, which assumes linearity) in our analyses.

In my opinion, deriving empirical unit hydrograph for each catchment at least for a sizable number of single-pulsed events will shed light to the results very significantly. As far as I can tell, the authors have the data to do this. Solving this inverse problem is tricky but doable, given that the authors have high-resolution rainfall and streamflow data. Such analysis would also be entirely in line with the data-driven approach.
AR: Unit hydrograph and similar empirical models make simplifying assumptions, including spatially homogeneous rainfall and fixed rainfall-catchment response relationships, that are not compatible with some of the objectives of the paper. Our analyses aim to identify if and under what conditions such assumptions are realistic. In fact, they show that for the majority of storms, storm characteristics and catchment response are far too complex to be modelled through these simplified

relationships. Consequently, all the models could show us is poor fits for most of the storms; this, the data can tell us more straightforwardly.

1.1 Use of radar data (or lack thereof)
In my view, the authors over-rely on the RWD analysis which is basically a proxy for excess rainfall (or runoff depth)-weighted travel time to the outlet. Because it does not account for spatially-varying velocities, attenuation effects, storage effects, nonlinear effects and integration effects, I do not think it is very amenable to quantitative analysis other than perhaps using as an index to infer the general location of the precipitation core relative to the outlet. If that is the case, I strongly think that the authors are better off examining the radar rainfall data directly. They will show with great certainty where the heavy rainfall was and in which direction the storm was moving, etc. Similarly, I find the exposition on storm vs. catchment scale to be a rather roundabout way to deal with the issue. It would be quite straightforward to characterize the size of the heavy rain cores directly from the radar rainfall data.

AR: the reviewer's interpretation of the rainfall-weighted flow distance (RWD) analysis is partly correct: it represents rainfall-weighted distance (along the flowpath network), which equates to travel time only if mean flow velocities along the network are the same across the catchment and across events. The latter is an assumption often made in simplified, empirical hydrologic models. Instead, we analyse the integrated effect of varying flow velocities, attenuation and storage along the flowpath network on hydrological response. This tells us to what extent the position of the storm relative to the catchment determines flow peak and lag time. If we were to analyse rainfall data directly, as the reviewer suggests, we would lose the relation with the catchment characteristics that we aim to analyse.

1.2 Stormwater infrastructure
The authors acknowledge its existence, including dams, but it is completely unclear what they are and what impact they may have. Because the size of the storms that the authors are dealing with is small (the largest several events per year), one would expect potentially significant impact by the storm drain network. The impact by the dams and other detention structures would potentially be greater. Little of this, however, is explained or justified.

AR: The only quantitative information available to us about stormwater infrastructure in the Charlotte watershed is the number of dams, which is low for all 5 catchments (0, 1, 0, 5 and 8 for the smallest to largest catchment). Additionally, a study has been recently published by Bell et al (2016), that includes some additional information for 3 of the 5 catchments we studied. Based on this, they computed the percentage area of mitigated area by detention structures: 5.5, 5.8 and 3.2 % for Little Hope, Upper Briar and Upper Little Sugar, respectively. These numbers show that the impact of detention structures on hydrological response is likely to be very small.
The reviewer is right that this information is relevant; we added this reference and related information to the manuscript in section 3.1 (p.13).

1.3 Flowpath analysis
It is completely unclear how this was done. Is this meant to capture channel flows only or both channel and hillslope flows? In their lag time analysis, how did they account to spatially varying roughness/velocity? The nature of this analysis has large implications in interpretation of the results.

AR: The methodology of rainfall-weighted flow distance analysis has been used in multiple previous, well-cited publications by our group as well as by other groups (Smith et al (2002); Smith et al (2005); (Zoccatelli et al. (2011); Nikolopoulos et al. (2014); Emmanuel et al (2015)). It represents the position of a storm relative to the flowpath network and is used to analyse how storm position and movement influence hydrological response (flow peak and lag time).
Regarding the reviewer's question about lag time analysis: we derive lag times directly from the data, as explained in section 2.2.1. Hence, there is no need to make assumptions about flow velocities as one would do in an empirical hydrological model.

2 General lack of clarity and specificity

I find the manuscript a very difficult and frustrating read due to loose notations and very liberal use of certain expressions. I illustrate this using a couple of examples below. Hydrologic response – I am not sure exactly what the authors mean by this expression which is used numerous times throughout the manuscript. In this work, the authors deal with streamflow response at the catchment and subcatchment outlets only. Urban flooding is a concern not only along the main channels, for whose response the outlet flow is a reasonable descriptor, but also in all upstream areas. I was led to believe by the title that this study deals with the role of spatiotemporal variability of rainfall on urban flooding across scale but it is largely about catchment- and subcatchment-wide response to rainfall.

AR: we realise that the use of the term "rainfall spatial distribution" in the abstract may have been misleading and will replace this by "storm position, movement and scale", as indicated in our reply above. To our knowledge, the term hydrological response is commonly used to describe aspects of rainfall-response in hydrological systems, including peak flow, lag time, runoff ratio etc. The term flood response, then, is used refer to hydrological response to intense events, in the upper tail of the rainfall frequency distribution. To clarify this point, we have rephrased the text in the abstract and introduction, where we outline the context and purpose of our study, to "hydrological response at the (sub)catchment"

Variability - The authors introduce many different types of variability in the manuscript: spatial variability, temporal variability, catchment variability, flow variability, peak flow variability, lag time variability, variability in runoff ratio expressed in terms of CV, climatological variability and possibly more. Many of these expressions are, however, rather loosely defined or undefined. For example, by "climatological variability", I believe the authors mean event-to-event variability. Also, fractional coverage is part of spatial variability of rainfall. If the authors mean inner variability, i.e., variability of positive rainfall by "variability of rainfall", they should indicate as such. If CV is used to measure variability, the authors should clearly state of what quantity, if not the complete mathematical expression. Again, the numerous loose descriptions, definitions and notations (see below) make reading this manuscript rather frustrating in that one has to guess at what the authors may actually mean.

AR: we have used the term variability predominantly to refer to spatial variability of rainfall and to variability in frequency distributions, in terms of coefficient of variation or inter-quantile range (for distributions of values for peak flow, lag time and runoff ratio. This terminology is commonly used in the literature and we did not expect it to be a cause of confusion. We have screened the text and replaced the term "variability" by more specific terms like quantile range and CV, where appropriate.

3 Inconsistent and missing notations

There are many places where the notations are missing, inconsistent, if not incorrect, or confusing. For example, on page 9, $r$ and $r(t,x)$ are never defined. If they mean the same, this is an abuse of notation as the former is a variable and the latter is a function. Also, the usual notation would be $r(x,t)$, not $r(t,x)$. Neither is $DRw(t)$ defined. I do not see how $D(t)$ is a random variable that takes values from 0 to 1. According to Eqs.(8) and (9), if there is excess runoff at time $t$, $D(t)$ should be zero (assuming $r(t,x)$ denotes rainfall at time $t$ and flow path $x$). And yet, in Fig 5, RWD seems to be positive even when $r(t,x)$ is zero.

AR: Indeed, a definition of $r(t,x)$ , used in equation 9, is missing. Thanks for spotting the omission, we will correct this. $DRw(t)$ should be $D(t)$, this will be corrected. $D(t)$ is a distance value normalised by maximum flow distance and varies from 0 at the outlet  to 1 at the maximum flow distance, i.e. at the headwaters of the catchment, as explained in section 2.2.2. Since RWD is distance multiplied by weighted rainfall it is indeed zero when rainfall is zero. In figure 5, RWD is above zero only when rainfall intensity is above zero (it may be very low, but not zero).

4 Significance

There are 7 specific conclusions the authors draw from this work which are stated in the Summary and Conclusions Section as well as in the abstract. In my view, most of them are already well known and established. I suspect that most practicing hydrologists and water resources engineers, particularly in urban areas, would find them largely a restatement of what they already know and practice. For the last "unexpected" conclusion, the authors state "We find that urbanisation plays a minor role in explaining variability in peak flow and lag time in the five basins in Little Sugar Creek." It is not completely clear what is meant by "variability of peak flow and lag time" but, assuming the authors meant event-to-event variability, the above is explained by the following two observations. The first is that these are small catchments (_111.1 km2) and hence, when there is heavy rainfall, it is very likely rain over most or all of the catchment area. This greatly reduces the likelihood of impervious areas amplifying event-to-event variability in runoff generation as they will almost always generate runoff. The second is that, unlike pervious areas, impervious areas will run off essentially all rainfall. As such, there is little event-to-event variability to be expected over impervious areas in small catchments.

AR: We believe the conclusions are not quite as obvious as the reviewer suggests. A few examples to illustrate this:

- in conclusion 2: "Lowest peak flow variability is found for the most urbanised basin". In many previous studies it has been assumed that urbanisation leads to higher peak flow variability.

- in conclusion 5 and 6: the position and movement direction of a storm play a minor role in explaining variability in hydrological response compared to rainfall volume and peak intensity. This is contrary to previous studies, where storm position and movement have been found to influence flow peak and lag time, often based on very small data samples or theoretical modelling studies (see e.g. Ogden et al., 1995; Seo et al., 2012; Ruiz-Villanueva et al., 2012). It is important to recognise that this large set of field data challenges previous findings.

- last conclusion: contrary to what the reviewer states, our data show (figure 3) that the scale of a large fraction of the storms is (much) smaller than basin scale, especially for the larger basins (>10km2). Hence, one would expect variability in flow response for storms that are spatially concentrated over urban regions versus those that are concentrated over non-urban regions. Our data do not confirm this and we give possible explanations why the field data are probably showing a strongly smoothed signal.

Interactive comments on "The role of storm dynamics and scale in controlling urban flood response"

Reviewer 3:
This paper presents a thorough and well-presented empirical analysis of storm rainfall and runoff across a number of highly urban basins. It is perhaps overly ambitious in brining so many facets together in one paper, leading to some difficulty for the reader to seperate each of the analyses undertaken, but this is balanced by high quality analysis on a large number of high resolution flood events across 5 basins. The paper has 7 substantial conclusions, and each of them is based on a sound analysis of robust data. The language and presentation is overall good, and the paper is well presented.

Specific comments
The last two sentences of the abstract are unclear and unjustified – they can be improved easily.
AR: we have rephrased the last sentences of the abstract as follows:
"Unexpectedly, position of the storm relative to impervious cover within the basins had little effect on flow peaks. These findings show the importance of observation-based analysis in validating and improving our understanding of interactions between spatial distribution of rainfall and catchment variability."

The role of soil moisture has not been considered in the paper – can the authors comment and justify on why this has not been considered in their analyses.
AR: in a separate study by Zhou et al (2017) the influence of antecedent rainfall, as a measure for soil moisture content, on flood response was analysed for a larger group of catchments in this region. They did not find a significant impact and concluded that other factors are dominant in explaining flood response. The following text was added in section 1 (p. 4):
"Our study focuses on spatial storm characteristics and does not look at effects of time between rainfall events or the time since the last rainfall event and how that affects the flood peaks. In a recent study by Zhou et al. (2017) the effect of watershed wetness was investigated for the Charlotte region; they did not find a significant influence of antecedent rainfall on flood response."

Figure 2 and the data– are the event data normal and if not have they been normalized before statistical comparison between events. Also – for rainfall, are the rainfall events in fact not independent – and does this not affect the validity of any comparison between catchments if indeed what is being compared is essentially the same rainfall events that pass over them all? Which sites are significantly different in the plot?
AR: The data are not normally distributed, as one can see from the skewedness of the boxplots in figure 2. Data have been normalised by catchment area to compare between basins of different size. Table 2 shows the degree of overlap between storms for the different catchments: varying between 20% for the basins furthest apart (LHope and UBriar) to 69% for the upper and lower LSugar Creek basins. However, as we can see in figure 2, differences in degree of overlapping storms between basins do not result in more similar rainfall or flow patterns
We added the following text (p. 14-15): "As we can see in figure 2, a higher degree of overlapping storms between basins does not result in more similar rainfall or flow patterns: rainfall and flow characteristics are as similar or dissimilar for Upper compared to Lower LSugar Creek as they are for LHope and UBriar or other sets of non-overlapping basins. Even if flood events in different catchments are generated by the same rainfall events, the characteristics of the rainfall as it affects the catchments is very different."

Figure 5 – what is the z axis scale on line 2 – 0-1%? I assume it means 0-100%. Also one of the plots then exceeds 100% in the graphic.

AR: thanks for pointing this out, the scale should be 0-1, not %. The peak value that seemed to exceed 1 is an artefact caused by the line thickness. We have adjusted the figure (z-scale and line thickness).

The general layout is difficult to follow as tables are referenced well before they are placed in the document – which can make the paper hard to follow – can this be improved in the final manuscript (e.g. Table 4).
Tables have been placed closer to where they are referenced.

I'm confused with Tables 3 and 4 and how they are used in the conclusions – please address the following points.
1.  In table 3 you state associated p values are set out, but I see no actual reported p values, only asterix to indicate a p value that is significant, here at 5%.
2.  Next in table 4 the significant correlations are in bold, rather than asterix being used. In both it seems Spearman rank correlation and significance - see Table 3 where LLSugar has a 0.25* for Tlag-RWD(ih), while LHope has 0.26 - are not related.
3.  First in conclusion 4 its stated that dynamics of rainfall coverage are important drivers of rainfall variability – with spearman ranks values exceeding 0.8 for the five basins –from where is this data taken or reported in the paper – what table reports this?
4.  Next in conclusion 4 you note maximum rainfall coverage (storm core?) is significantly and positively correlated with peak flow for two of the five basins (smallest and largest), with values of 0.33, and not significant correlation in the others. Again I cannot seem to link this reporting to the results in text or table. The only 0.33 reported is for UBriar in table 4 and also referred to in the text.
AR: We hereby reply to each of the 4 points mentioned by the reviewer; thanks for pointing out the inconsistencies between tables and conclusions:
1.-2.    Tables 3 and 4 have been merged into a single table, significance at the 5% level is indicated by asterix symbols.
3.  Spearman rank correlation values for first-order differences in rainfall coverage versus rainfall intensity are reported in section 3.2 (just above heading of section 3.3.). The conclusion was rephrased to make the connection with the text more clear.
4. We agree this conclusion was not clear. We have split conclusion 4 into two separate conclusions and revised the text substantially.

I feel conclusion 7 is interesting and warrants further discussion or possible explanation – as urbanisation more than doubles in some catchments and the general consensus is more urbanisation equals more runoff and higher peak flows. This should also include some caveat regarding the fact storm water infrastructure was not included.
AR: we have rephrased conclusion 7 to more clearly reflect the conclusion we draw from our analyses: "
[revised manuscript text omitted]